# Local Correlation-based Transition Models for High-Reynolds-Number Wind Turbine Airfoils

Yong Su Jung[1], Ganesh Vijayakumar[2], Shreyas Ananthan[2], and James Baeder[3]

[1]Pusan National University, Pusan, Republic of Korea
[2]National Renewable Energy Laboratory, Golden, CO
[3]University of Maryland, College Park, MD

**Correspondence:** Yong Su Jung (yongsu.jung@pusan.ac.kr)

**Abstract.** Modern wind-turbine airfoil design requires robust performance predictions for varying thicknesses, shapes, and appropriate Reynolds numbers. The airfoils of current large offshore wind turbines operate with chord-based Reynolds numbers in the range of 3-15 million. Turbulence transition in the airfoil boundary layer is known to play an important role in the aerodynamics of these airfoils near the design operating point. While the lack of prediction of lift stall through Reynold-averaged Navier-Stokes (RANS) computational fluid dynamics (CFD) is well-known, airfoil design using CFD requires the accurate prediction of the glide ratio ($L/D$) in the linear portion of the lift polar. The prediction of the drag bucket and the glide ratio is greatly affected by the choice of the transition model in RANS-CFD of airfoils. We present the performance of two existing local correlation-based transition models—one-equation ($\gamma-$SA) and two-equation model ($\gamma - \overline{Re_{\theta t}}-$SA) coupled with the Spalart-Allmaras (SA) RANS turbulence model—for offshore wind-turbine airfoils operating at a high Reynolds number. We compare the predictions of the two transition models with available experimental and CFD data in the literature in the Reynolds number range of 3-15 million including the AVATAR project measurements of the DU00-W-212 airfoil. Both transition models predict a larger $L/D$ compared to fully turbulent results at all Reynolds numbers. The two models exhibit similar behavior at Reynolds numbers around 3 million. However, at higher Reynolds numbers, the one-equation model fails to predict the natural transition behavior due to early transition onset. The two-equation transition model predicts the aerodynamic coefficients for airfoils of various thickness at higher Reynolds numbers up to 15 million more accurately compared to the one-equation model. As a result, the two-equation model predictions are more comparable to the predictions from $e^N$ transition model. However, a limitation of this model is observed at very high Reynolds numbers of around 12-15 million where the predictions are very sensitive to the inflow turbulent intensity. The combination of the two-equation transition model coupled with the Spalart-Allmaras (SA) RANS turbulence model is a good method for performance prediction of modern wind-turbine airfoils using CFD.

# 1 Introduction

The aerodynamic design of increasingly large rotors (Veers et al., 2019) to satisfy the world's wind energy needs relies on robust and accurate performance predictions at all operating conditions. The airfoils of current large wind turbines operate at chord-based Reynolds numbers of 3-15 million, as shown in Fig. 1. Laminar-turbulent boundary layer transition is a complex phenomenon that affects the aerodynamics of airfoil boundary layers near the design operating point. Reynolds-averaged Navier-Stokes (RANS) modeling using computational fluid dynamics (CFD) is a common high-fidelity modeling tool used for airfoil design. Typical RANS-CFD solvers are augmented with transition models to improve accuracy of aerodynamic predictions of airfoils.

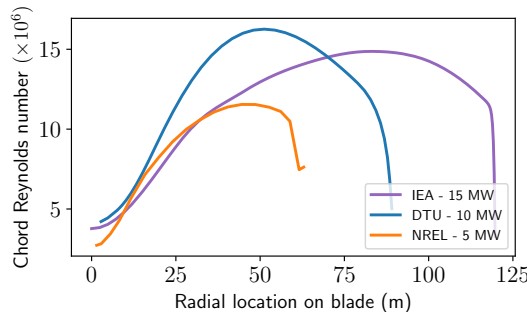

**Figure 1.** Variation of chord Reynolds number and airfoil thickness along the blade span for three modern, open source, MW-scale turbines: NREL 5-MW (Jonkman et al., 2009), DTU 10-MW (Bak et al., 2013), and IEA 15-MW (Gaertner et al., 2020)

This work is part of a project to develop a machine-learning inverse-design capability for three-dimensional (3D) aerodynamic design of wind turbine rotors. In the first phase of our project, we focus on inverse-design of two-dimensional (2D) airfoils. Our goal is to develop a robust 2D airfoil capability with the appropriate transition model that can accurately predict the performance of airfoils of various thicknesses and shapes at different operating conditions to generate reliable training data for the machine-learning process. It is well-known that 2D RANS-CFD does not accurately capture the stall behavior of airfoils because it is an unsteady 3D phenomenon (Ceyhan et al., 2017b). Airfoils are typically designed to operate inside a range of angles of attack for maximum performance away from stall in the linear portion of the lift curve. Hence, the generation of training data for airfoil-design purposes requires the accurate prediction of the glide ratio inside the design range of angles of attack. The variation of the glide ratio near the design points is highly sensitive to the boundary layer transition onset location.

Transition modeling and simulation are divided into analytical models based on stability theory and statistical models. The $e^N$ model is a popular analytical transition model based on the linear stability theory. However, the application of the $e^N$ method within a conventional RANS framework that runs on massively parallel computers is difficult. This is because it involves non-local search and line integration operations for boundary layer quantities (e.g. displacement/momentum thickness and shape factor). Also, additional efforts in communications between $e^N$ and RANS methods are required (Sheng, 2017). In wind turbine applications, the $e^N$ method has been used in either 2D RANS flow solvers or a low fidelity XFOIL code (Sorensen

et al., 2016; Ceyhan et al., 2017b). However, much more complex infrastructure is required in coupling it with a full 3D RANS-CFD method.

Statistical models like local correlation-based transition models (LCTMs) that solve prognostic transport equations for transition variables are more suitable for use with RANS-CFD models. Two major LCTMs are the two-equation $\gamma - \overline{Re_{\theta t}}$ model developed by Langtry and Menter (2009), and the simplified one-equation $\gamma$ model by Menter et al. (2015). The one-equation model is preferable for wind-turbine modeling, as it satisfies Galilean invariance, a requirement for application to rotating physical systems. These transition models were originally developed to be coupled with the shear stress transport ($k$-$\omega$-SST) turbulence model that is widely used in the wind turbine community. However, different versions of LCTM coupled to the one-equation SA turbulence model have also been developed (Medida, 2014; Wang and Sheng, 2014; Nichols, 2019). The SA turbulence model has advantages of robustness, reliability, and lower computational cost than the ($k$-$\omega$-SST) model. The LCTM-SA models have been successfully applied to a wide range of aerospace problems including rotorcraft.

Applications of RANS-CFD to wind turbine modeling have mostly focused on using the $k$-$\omega$-SST turbulence model coupled to the LCTM or the $e^{\mathrm{N}}$-based transition model (Sorensen et al., 2016; Ceyhan et al., 2017b). Sorensen et al. (2014) showed that the two-equation $\gamma - \overline{Re_{\theta t}}$ transition model fails to correctly predict natural transition behaviors at high Reynolds numbers compared to the $e^{\mathrm{N}}$-based model. Two out of the four codes in the blind-test campaign (Ceyhan et al., 2017b) to predict the performance of the DU00-W212 airfoil using AVATAR data (Ceyhan et al., 2017a) also used the $e^{\mathrm{N}}$-based model. Hence, the above studies together show the superiority of the $e^{\mathrm{N}}$-based method over LCTM for predicting natural transition behavior for high-Reynolds-number flows. However, there is a lack of studies using the LCTM coupled with the SA turbulence model for wind turbine applications.

In this paper, we quantify the performance of both one-equation and two-equation LCTM coupled with the SA turbulence model in simulating wind turbine airfoils at a wide range of Reynolds numbers. We compare our simulation results to not only experiments but also the other predictions using different transition models (e.g. $e^{\mathrm{N}}$-based). First, the formulations of correlation-based transition models are presented briefly for completeness in Section 2. Then, we show validation and code comparison results for the SA turbulence model using fully-turbulent approximation in Section 3. Section 4 analyzes the differences between the predictions from the one-equation and two-equation transition models through comparison to experimental and other reference data in the literature. This includes the comparison to the measurements from the AVATAR project (Ceyhan et al., 2017a) on the DU00-W-212 airfoil at Reynolds numbers 3-15 million. We then compare the predictions of the two LCTMs for airfoils from three modern, open source, MW-scale wind turbines, NREL 5 MW (Jonkman et al., 2009), DTU 10 MW (Bak et al., 2013), and IEA 15 MW (Gaertner et al., 2020). Our simulation results are compared with available reference data from experiments and/or other simulations in the literature. Finally, we conclude with a discussion of the transition models in RANS-CFD solvers for airfoil design in modern wind turbines.

## 2 Methodology

### 2.1 Reynolds-Averaged Navier-Stokes Solver

The Hamiltonian solver (HAM2D) is a Reynolds-averaged Navier-Stokes (RANS) flow solver that was developed at the University of Maryland (Jung and Baeder, 2019). This is a parallel solver for the solution of the two-dimensional compressible Navier-Stokes equations on unstructured meshes using finite volume formulation. A fifth-order weighted essentially non-oscillatory (WENO) scheme is used for spatial reconstruction and Roe's approximate Riemann solver is used to compute inviscid fluxes. Viscous fluxes are calculated using second-order central differencing. For the steady-state solution, the preconditioned generalized minimum residual (GMRES) is used as implicit time-integration method. The turbulent boundary layer is modeled using one-equation SA model. Both the one-equation $\gamma-$SA transition model (Lee and Baeder, 2021) and the two-equation $\gamma - \overline{Re_{\theta t}}-$SA transition model (Medida, 2014) have been coupled with the SA turbulence model to predict boundary layer transition if necessary. In this study, the incompressible flow condition was approximated in the compressible solver using a freestream Mach number of 0.1. The Reynolds number based on chord length and angle of attack was adjusted for test flow conditions.

Our in-house automated airfoil mesh generation was used for various test airfoils, which is designed to require relatively few control inputs from airfoil coordinates (Costenoble et al., 2018). For efficient meshing, the surface point distribution (clustering/stretching) is based on local sharp corners or different surface curvatures along the airfoil. An O-type grid is used to allow for a finite-thickness trailing edge. A strand-/advancing-front-based method is used to generate a body-fitted mesh around the airfoil, and the triangle elements are used to extend the domain to the outer boundary. All triangular elements are transformed to obtain a pure quadrilateral mesh, which is required by the flow solver. In previous studies, the current flow solver and automated mesh generation have been validated through various canonical problems (Costenoble et al., 2017, 2018; Jung and Baeder, 2019). For the simulations in this paper, the number of nodes in the wrap-around direction was fixed as 400 points, as determined through a grid convergence study (Appendix B), and the initial wall-normal spacing was varied according to the test Reynolds number, such that $y^+ = 1$. The outer boundary was placed 300 chord lengths away from the airfoil where the far-field boundary condition was imposed. Figure 2 shows the mesh generated using this method around the DU93-W-210LM airfoil at a Reynolds number of $9 \times 10^6$ as an example. The convergence of the residuals and the aerodynamic coefficients with solver iterations is shown in Appendix C.

### 2.2 Two-Equation Laminar-Turbulent Boundary Layer Transition Model

The two-equation LCTM model used in this study is the $\gamma - \overline{Re_{\theta t}}-$SA model, also known as the Medida-Baeder transition model. A brief description of this transition model is presented in this paper and a detailed description can be found in the previous studies (Medida, 2014; Jung and Baeder, 2019). This transition model can predict natural transition, separation-induced transition, and bypass transition and has been validated through various canonical problems. The transition model uses the concept of intermittency, $\gamma$, in order to trigger transition locally. The intermittency is a scalar transport variable that varies

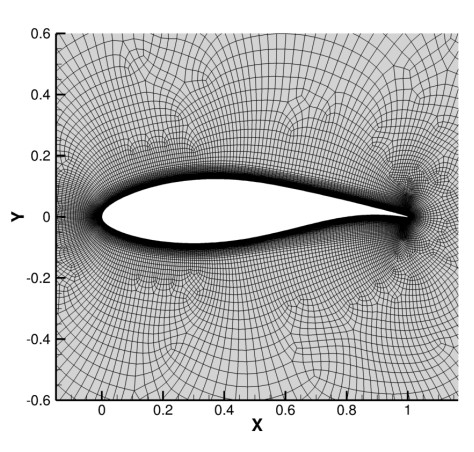
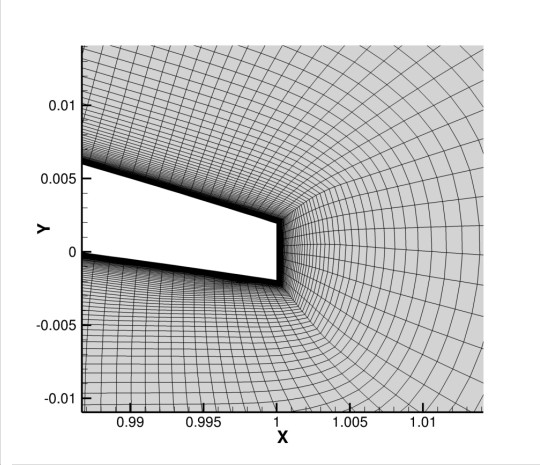

(a) Mesh around the DU93-W-210LM airfoil  (b) Mesh at the finite-thickness trailing edge

**Figure 2.** Example of a computational mesh for the DU93-W-210LM airfoil

between 0 (pure laminar) and 1 (pure turbulent). The transport equation for the intermittency is given by

$$\frac{D(\rho\gamma)}{Dt} = P_\gamma - D_\gamma + \frac{\partial}{\partial x_j}\left[(\mu+\mu_t)\frac{\partial\gamma}{\partial x_j}\right], \tag{1}$$

where $P_\gamma$ and $D_\gamma$ denote the production and destruction term, respectively.

The transport equation for transition momentum thickness Reynolds number, $\overline{Re_{\theta t}}$, is used to account for history effects of pressure gradient on determining the onset of transition. This equation is given by

$$\frac{D\left(\rho\overline{Re_{\theta t}}\right)}{Dt} = P_{\theta t} + \frac{\partial}{\partial x_j}\left[2.0(\mu+\mu_t)\frac{\partial\overline{Re_{\theta t}}}{\partial x_j}\right], \tag{2}$$

where $P_{\theta t} = 0.03\frac{\rho}{t}(Re_{\theta t} - \overline{Re_{\theta_t}})(1.0 - F_{\theta t})$.

     Once the distribution of $\overline{Re_{\theta_t}}$ in the computational domain is solved, the critical momentum thickness Reynolds number

is obtained through $Re_{\theta c} = 0.62 \cdot \overline{Re_{\theta_t}}$. Then, the intermittency production can be triggered based on the ratio of the local vorticity Reynolds number, $Re_v$, to the $Re_{\theta c}$. For the production term, the transition onset momentum thickness Reynolds number, $Re_{\theta_t}$, is computed through the empirical correlations in an iterative manner, which are functions of the streamwise pressure gradient parameter, $\lambda_\theta$, and the inflow turbulent intensity. $\lambda_\theta$ is defined as

$$\lambda_\theta = \frac{\rho\theta^2}{\mu}\frac{dU}{ds}, \tag{3}$$

$$\frac{dU}{ds} = \frac{u}{U}\frac{dU}{dx} + \frac{v}{U}\frac{dU}{du} + \frac{w}{U}\frac{dU}{dz}, \tag{4}$$

where $U = \sqrt{u^2 + v^2 + w^2}$.

When this transition model is coupled with the SA turbulence model, the intermittency is used to control only the production term of the transported variable, $\tilde{\nu}$, as

$$\frac{D\tilde{\nu}}{Dt} = \gamma P_{\tilde{\nu}} - D_{\tilde{\nu}} + \frac{1}{\sigma}\left[\nabla \cdot ((\nu + \tilde{\nu})\nabla\tilde{\nu}) + c_{b2}(\nabla\tilde{\nu})^2\right]. \tag{5}$$

Details of the current implementation of the transition model compared to $\gamma - \overline{Re_{\theta t}}$ model by Langtry and Menter (2009) are shown in the previous study (Medida, 2014; Jung and Baeder, 2019)

## 2.3  One-Equation Laminar-Turbulent Boundary Layer Transition Model

The one-equation transition model first proposed by Menter et al. (2015) uses only the intermittency variable, $\gamma$: hence, only the transport equation for the intermittency is required as shown in Eq 1. Both production and destruction terms for the intermittency are different compared to $\gamma - \overline{Re_{\theta t}}$ model. The transport equation for $\overline{Re_{\theta t}}$ is replaced with the empirical-based formulation as follows for obtaining $Re_{\theta c}$.

$$Re_{\theta c}(Tu_L, \lambda_{\theta L}) = C_{TU1} + C_{TU2}exp[-C_{TU3}Tu_L F_{PG}]. \tag{6}$$

Our implementation of the one-equation model uses modified coefficients of $C_{TU1}$, $C_{TU2}$, and $C_{TU3}$ compared to that by Menter et al. (2015), which gives better correlation with the experiments than the original values, as shown by Colonia et al. (2016). The modified values of the constants are

$$C_{TU1} = 163.0,\ C_{TU2} = 1002.25,\ C_{TU3} = 1.0. \tag{7}$$

In the $Re_{\theta c}$ formulation, $F_{PG}$ is introduced to sensitize the transition onset to the streamwise pressure gradient. The pressure gradient parameter, $\lambda_\theta$, in Eq. 4 is approximated as $\lambda_{\theta L}$ in the model; thus it becomes only a function of wall normal direction velocity and coordinate in addition to the kinematic viscosity, $\nu$.

$$\lambda_{\theta L} = -7.57 \cdot 10^{-3}\frac{dV}{dy}\frac{d_w^2}{\nu} + 0.0128, \tag{8}$$

$$\lambda_{\theta L} = \min(\max(\lambda_{\theta L}, -1.0), 1.0), \tag{9}$$

$$\frac{dV}{dy} = \Delta\left(\boldsymbol{n} \cdot \boldsymbol{V}\right) \cdot \boldsymbol{n}, \tag{10}$$

where $d_w$ is the wall distance.

The one-equation transition model was coupled with the SA turbulence model first by Nichols (2019) using the equations as follows:

$$\frac{D\tilde{\nu}}{Dt} = \tilde{P}_{\tilde{\nu}} + P_{\tilde{\nu}}^{lim} - \tilde{D}_{\tilde{\nu}} + \frac{1}{\sigma}\left[\nabla \cdot ((\nu + \tilde{\nu})\nabla\tilde{\nu}) + c_{b2}(\nabla\tilde{\nu})^2\right], \tag{11}$$

$$\tilde{P}_{\tilde{\nu}} = \gamma_s P_{\tilde{\nu}}, \tag{12}$$

$$\tilde{D}_{\tilde{\nu}} = \max\left(\gamma_s, 0.1\right) D_{\tilde{\nu}}. \tag{13}$$

It should be noted that the intermittency is used to control both the production and destruction terms of the SA model unlike the two-equation transition model. Nichols (2019) also defined a re-scaled transition variable, $\gamma_s$, which goes from zero at the wall to one in turbulent regions of the flow as below. This is because the SA model requires the production source term to go to zero in laminar regions of the flow.

$$\gamma_s = \frac{\min(\gamma, 1) - 1/c_{e2}}{1 - 1/c_{e2}}, \tag{14}$$

$$\gamma_s = \max\left[\min\left(\gamma_s, 1\right), 0\right]. \tag{15}$$

In addition, $P_{\tilde{\nu}}^{lim}$ as an additional production term was proposed to ensure the generation of turbulent kinetic energy at the transition point for low free-stream turbulence intensity levels. Finally, for the local turbulence intensity computation, $Tu_L$, turbulent kinetic energy, $k$, and specific dissipation, $\omega$, variables were replaced as shown in the equations below because these variables are not available in the SA turbulence model.

$$Tu_L = min\left(100\frac{\sqrt{2k/3}}{\omega d_w}\right), \tag{16}$$

$$\omega = S/0.3, \tag{17}$$

$$k = \frac{\mu_t \omega}{\rho}. \tag{18}$$

where $S$ is the strain rate magnitude.

Based on the formulation for SA model by Nichols (2019), Lee and Baeder (2021) employed a constant freestream turbulence intensity assumption in the entire flow field, which is valid for external aerodynamic flows. An input turbulence intensity is a measured value from an experiment. The modified formulation was validated through canonical problems in both two and three dimensions (Lee and Baeder, 2021). Therefore, in the current study, we used the same formulation as the work by Lee and Baeder (2021) for the one-equation transition model.

## 3 Validation of turbulence model

In this section, we show code-to-code comparisons and validation studies for the SA turbulence model using our solver HAM2D. The performance of the SA turbulence model using the current solver and mesh-generation approach has been validated in previous studies (Jung et al., 2017; Jung and Baeder, 2019; Jung, 2019) through various test cases from NASA Turbulence Modeling Resource (TMR, 2017). The test cases include a 2D zero pressure gradient flat plate, 2D bump-in-channel, and NACA 0012 airfoil. The case of turbulent flow past a NACA0012 airfoil is shown in this paper as an example. The flow condition is a free-stream Mach number of 0.15, a Reynolds number of 6 million, and three angles of attack (0, 10, 15°). A structured airfoil C-type mesh ($897 \times 257$) provided by the TMR website is used for the current simulation. Table 1 shows the comparison of the lift and drag coefficients of the NACA0012 airfoil using the SA turbulence model. The force coefficients predicted by HAM2D are comparable to the results predicted by well-established legacy codes.

**Table 1.** Comparison of lift and drag coefficient for the NACA0012 airfoil using the SA turbulence model against other implementations in NASA-TMR (TMR, 2017).

| Codes | $C_l$ ($\alpha = 0°$) | $C_l$ ($\alpha = 10°$) | $C_l$ ($\alpha = 15°$) | $C_d$ ($\alpha = 0°$) | $C_d$ ($\alpha = 10°$) | $C_d$ ($\alpha = 15°$) |
|---|---|---|---|---|---|---|
| CFL3D | approx 0 | 1.0909 | 1.5461 | 0.00819 | 0.01231 | 0.02124 |
| FUN3D | approx 0 | 1.0983 | 1.5547 | 0.00812 | 0.01242 | 0.02159 |
| TURNS | approx 0 | 1.1000 | 1.5642 | 0.00830 | 0.01230 | 0.02140 |
| mStrand | approx 0 | 1.0967 | 1.5621 | 0.00804 | 0.01251 | 0.02195 |
| **HAM2D** | approx 0 | 1.0907 | 1.5459 | 0.00812 | 0.01232 | 0.02127 |

For wind-turbine airfoils, we compare our predictions using the SA model with those from EllipSys2D for FFA-301 and FFA-360GF airfoils using the $k$-$\omega$-SST model at a Reynolds number of 10 million under a fully turbulent flow assumption (Bak et al., 2013). Both predictions used enough fine meshes for the fully turbulent flow simulation, thus there is minor mesh dependency on both predictions. Also, both simulations neglected compressibility because EllipSys2D is a incompressible solver.

FFA-360GF is a very thick airfoil with $t/c_{\max} = 36\%$ and a gurney flap. Figure 3 compares the lift coefficient, drag coefficient, and lift-to-drag ratio as a function of angle of attack from each simulation. Both predictions show very good agreement in the drag coefficient and lift-to-drag ratio over the test angles of attack from -4° to 20°. Also, the linear portion of the lift polar is well matched between the predictions. This shows that the one-equation SA model provides similar performance compared to the two-equation $k$-$\omega$-SST model under fully turbulent flow conditions.

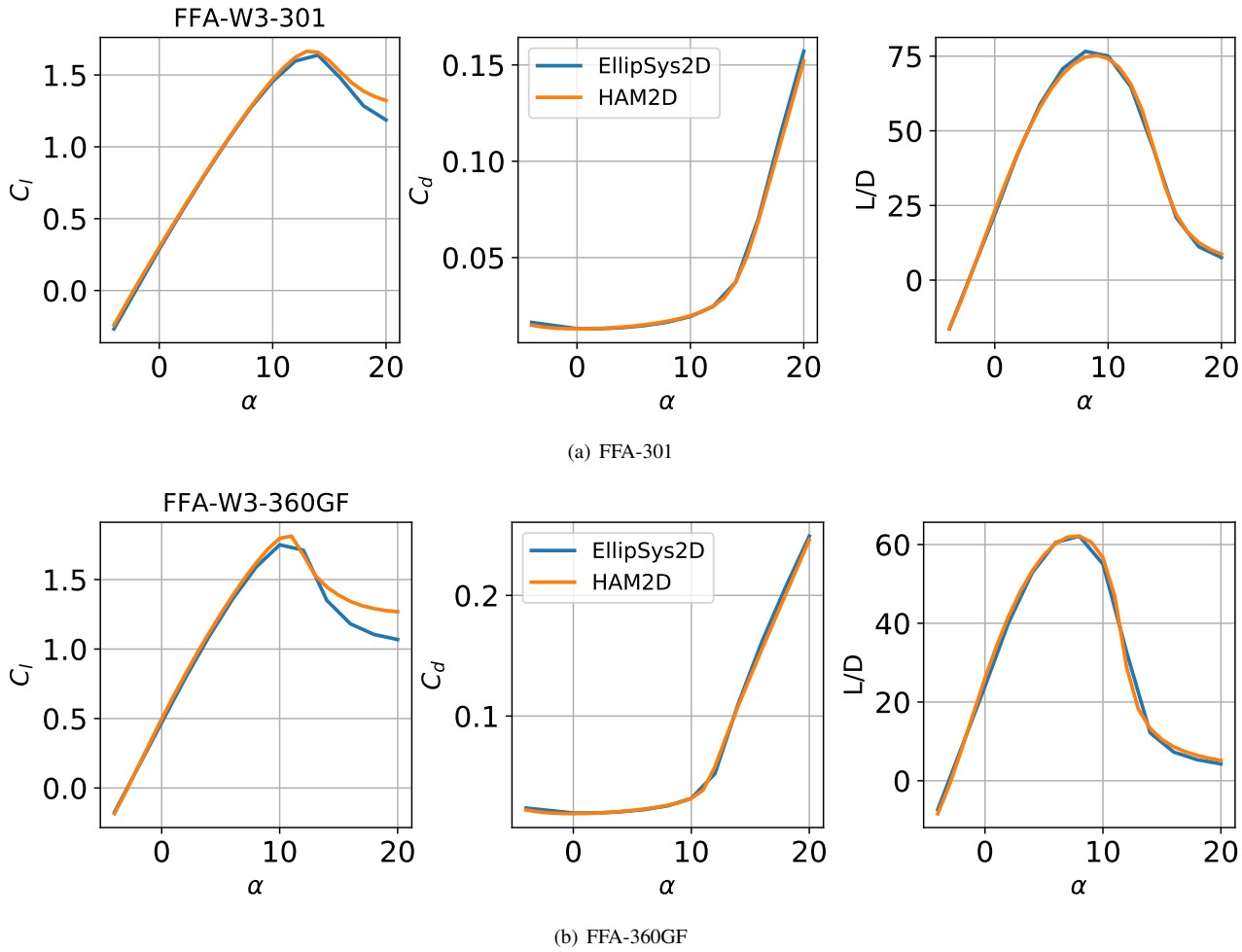

(a) FFA-301

(b) FFA-360GF

**Figure 3.** Comparison of aerodynamic performance predicted by HAM2D using the SA turbulence model with that from EllipSys2D using the $k$-$\omega$-SST turbulence model (Bak et al., 2013) under fully turbulent flow assumptions at a Reynolds number of 10 million.

# 4 Results: Transition modeling

We compare the aerodynamic load predictions from the one-equation and two-equation transition models on airfoils from modern wind turbines with available reference data from experiments and/or other simulation results in the literature. First we consider the S809 airfoil at a low Reynolds number of 2 million. Then, we compare the performance of the two transition models on the DU00-W-212 airfoil for which wind tunnel measurements are available through the AVATAR project (Ceyhan et al., 2017a) at Reynolds numbers of 3-15 million. We compare the airload with measurements in both fully turbulent and free transition conditions. The effect of the choice of transition model on the prediction of the transition onset location is analyzed. Finally, the sensitivity of freestream turbulent intensity on airload predictions using two-equation transition model is shown.

Next, we evaluate the effect of the transition model on other airfoils from three modern, open-source MW-scale wind turbines, NREL 5 MW (Jonkman et al., 2009), DTU 10 MW (Bak et al., 2013), and IEA 15 MW (Gaertner et al., 2020). To improve the readability, we show the comparison of the prediction from the two transition models for three representative airfoils (DU91-W2-250 and NACA64-618 airfoils from NREL 5MW and FFA-W3-301 airfoil from DTU 10 MW/IEA 15 MW) in this section and the rest in Appendices A1 and A2.

## 4.1 S809 airfoil

The two transition models considered in this study are evaluated through the S809 airfoil used in the NREL Phase VI wind turbine (Hand et al., 2001). We show validation of the aerodynamic performance prediction against experimental data (Somers, 1997) as well as previous simulation results using NASA's OVERFLOW code from Coder (2019) using SA-neg turbulence model with AFT2019 transition model. AFT2019 transition model was developed based on linear stability theory, which is also widely used in aerospace problems. It solves two transport equations for amplification factor and intermittency. We also compare our results with the other predictions using the two-equation transition model in OVERFLOW by Hall (2018).

The test flow condition parameters are a free-stream Mach number of 0.1, Reynolds number of 2 million based on chord length, and a free-stream turbulence intensity of 0.05%. We use the medium-resolution reference structured C-grid from the 2018 transition modeling workshop (Hall, 2018) with dimensions of $705 \times 87$ including 513 points on the surface and 97 points in the wake.

An angle-of-attack sweep was conducted from -8° to 15°. Figure 4 compares the lift polar, drag polar, and the transition location of the predictions from the fully-turbulent and free-transition simulations using both transition models against experimental data and other simulation results (Coder, 2019; Hall, 2018).

Figure 4 (a) shows that all simulations predict the lift coefficient well in the linear region of the lift polar. In detail, slightly higher lift coefficients from the untripped experiment than the tripped one are captured using either one- or two-equation model. Otherwise, the predictions significantly overpredict the maximum lift coefficient due to the known limitations of 2D CFD-RANS. Figure. 4 (b) shows that the drag predictions from HAM2D using the fully turbulent approximation are in excellent agreement with the OVERFLOW simulation results (Coder, 2019) at the same flow condition over the full range of angle of attack while showing a slight underprediction in the drag bucket compared to the tripped boundary layer experimental data.

The HAM2D predictions using both transition models show a similar underprediction inside the drag bucket while having excellent agreement with results from the AFT2019 model (Coder, 2019). The two-equation transition model predicts a slightly lower drag than the one-equation model inside the drag bucket and also shows an earlier departure from the drag bucket near a lift coefficient of 0.6 similar to the results using two-equation model from Hall (2018).

Figure 4 (c) compares the transition onset location, $X_T$, predicted by the current transition models with experimental data
on both the upper and lower sides of the airfoil. The transition onset location was determined by picking up the point in the middle of a sharp increase in the intermittency on the surface. As the angle of attack increases, the transition point on the upper surface moves to the stagnation point due to an increasing adverse pressure gradient. On the other hand, the transition onset on the lower surface moves downstream due to an increasing favorable pressure gradient with an increasing angle of attack. The transition occurs due to a short and intense laminar separation bubble on both sides of the airfoil (separation-induced
transition). Overall, the transition onset locations predicted by both transition models match well with the experimental data. The sharp movement on the upper surface at the 6° angle of attack was well captured by the two-equation model. However, this movement was predicted at 8° from the one equation model. This difference in the onset locations explains the earlier departure from the drag bucket using the two-equation model compared to the one-equation or the AFT2019 model.

## 4.2   DU00-W-212 airfoil - AVATAR

The AVATAR project from the European Union focused on aerodynamics of large rotors (Ceyhan et al., 2017a). The aerodynamic measurements on the DU00-W-212 airfoil from wind-tunnel experiments at conditions similar to those of 10 MW+ turbines were made publicly available through this project by Ceyhan et al. (2017a). We compare the effect of the one-equation and two-equation transition models against this data set as well as the results from the blind-test study by Ceyhan et al. (2017b) at Reynolds numbers of 3, 6, 9, 12, and 15 million. In the experiment, the lift and pitching moment coefficients were calculated
using pressure taps along the airfoil and the drag coefficient was calculated from the flow loss of momentum using the wake rake. It should be noted that the drag measurement can be inaccurate at post-stall region due to the nature of 3D flows which were measured using the wake rake at a fixed span location. Also, in the experiment, three different turbulent intensity levels were measured at the model location in different periods. Thus, we also study the sensitivity of the airload predictions to the inflow turbulence intensity level through the three different measured intensities shown in Table 2 and as performed by Ceyhan
et al. (2017b).

**Table 2.** Test matrix to analyze the effect of transition model at different Reynolds numbers and free-stream turbulence intensities (Ti) through comparison against experimental data for the DU00-W-212 airfoil from the AVATAR project (Ceyhan et al., 2017a). Three turbulence intensities (Ti1, Ti2, Ti3) are tested at each Reynolds number.

|  | Re=$3 \times 10^6$ | Re=$6 \times 10^6$ | Re=$9 \times 10^6$ | Re=$12 \times 10^6$ | Re=$15 \times 10^6$ |
|---|---|---|---|---|---|
| Ti[%] | Ti1=0.5129 | Ti1=0.8058 | Ti1=1.1877 | Ti1=2.2790 | Ti1=2.3944 |
|  | Ti2=0.3200 | Ti2=0.4600 | Ti2=0.4500 | Ti2=0.5100 | Ti2=0.5500 |
|  | Ti3=0.0864 | Ti3=0.1988 | Ti3=0.2448 | Ti3=0.3015 | Ti3=0.3346 |

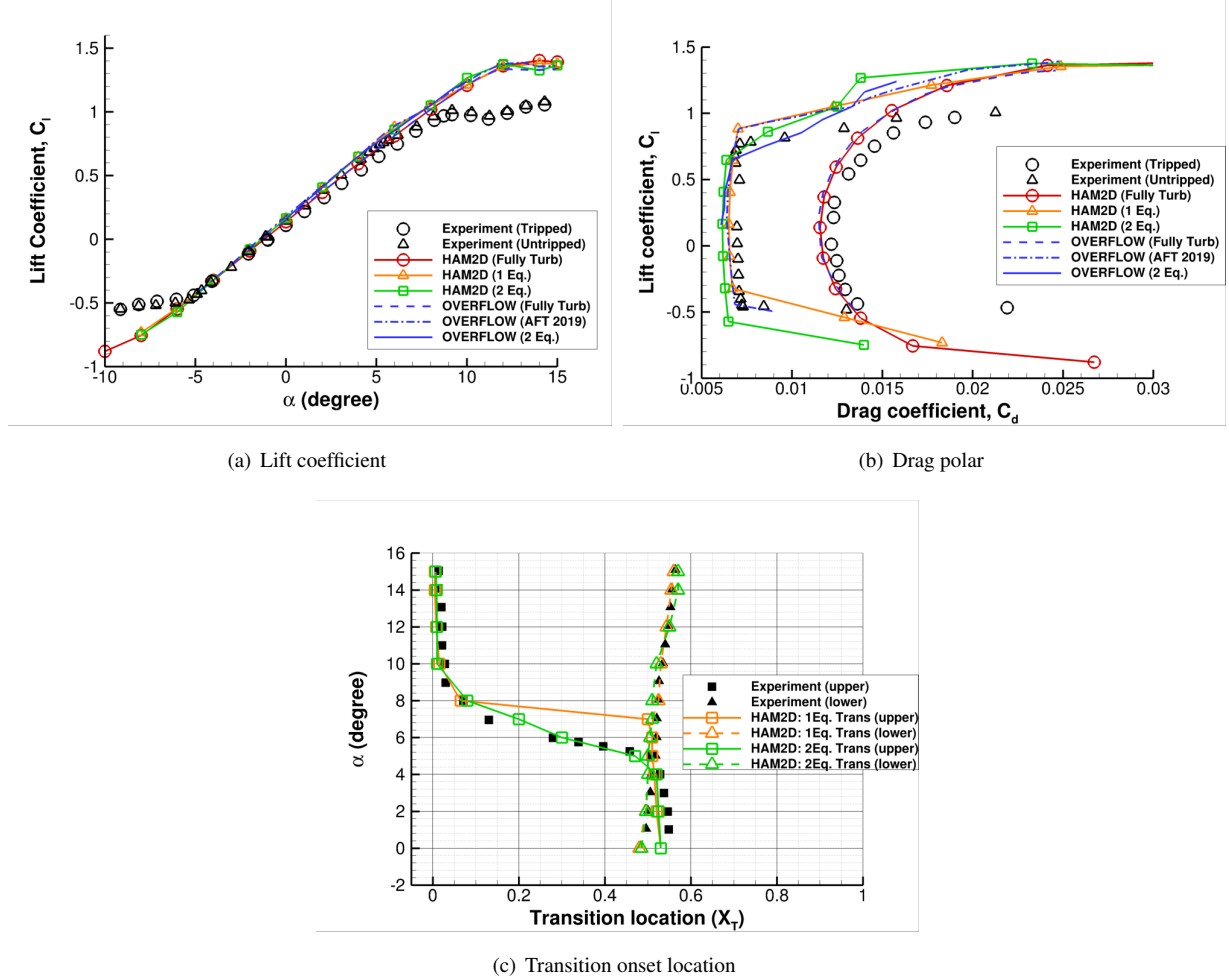

(a) Lift coefficient

(b) Drag polar

(c) Transition onset location

**Figure 4.** Comparison of (a) lift polar, (b) drag polar, and (c) transition onset location for the S809 airfoil at $Re = 2 \times 10^6$ predicted by HAM2D using a fully turbulent flow approximation and one-equation and two-equation transition models with experimental data (Somers, 1997). Also shown are reference predictions using the NASA-OVERFLOW solver using fully turbulent flow approximation and AFT2019 transition model (Coder, 2019), and the two-equation transition model (Hall, 2018).

The computational grid for the DU00-W-212 airfoil was generated using the automated mesh generation procedure described in Section 2. It has 500 points in the wrap-around direction and the initial wall-normal spacing of $1.8 \times 10^{-6}$ chord ($y^+$=1), which results in a grid with a resolution comparable to the meshes used by Ceyhan et al. (2017b).

We performed fully-turbulent and free-transition flow simulations with both transition models at the five different Reynolds numbers in Table 2 and the angles of attack ranging from -4° to 20°. Figure 5 shows the comparison of the lift-to-drag ratio and drag polar between the fully turbulent flow simulation results from HAM2D with the experimental data with the tripped boundary layer (Pires et al., 2016). Figure 5 (a) shows that HAM2D overpredicts the maximum lift-to-drag ratio, but the overall

trend from the experiment is captured well: the slope in the linear region increases as the Reynolds number increases and the maximum lift-to-drag ratio increases as the Reynolds number increases. In Fig. 5 (b), the minimum drag coefficients match well with experimental data in the linear region of the lift polar at all Reynolds numbers and decreases as the Reynolds number increases.

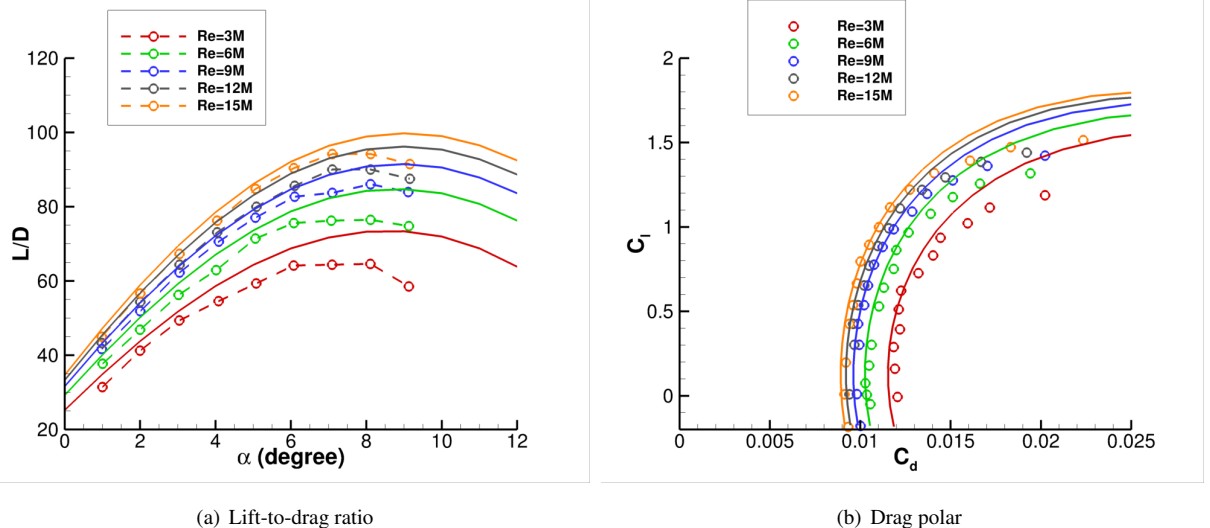

(a) Lift-to-drag ratio

(b) Drag polar

**Figure 5.** Comparison of (a) lift-to-drag ratio and (b) drag polar predicted by HAM2D using fully turbulent flow approximation for the DU00-W-212 airfoil against experimental data from Pires et al. (2016) with a tripped boundary layer.

Figure 6 compares the lift-to-drag (glide) ratio predicted by HAM2D using the one-equation and two-equation transition models with experimental data for the untripped boundary layer (Ceyhan et al., 2017a) and the other simulations from Pires et al. (2016). The other simulation results were obtained using the $k-\omega$-SST turbulence model coupled with different transition models: semi-empirical $e^{N}$ method by Drela-Giles for DTU-EllipSys, $e^{N}$ method combined with linear stability solver for Kiel-TAU, and Granville/Schlichting model (Ceyhan et al., 2017b) NTUA-MapFlow. The lowest turbulence intensity level (Ti3) was used at each Reynolds number for all the computations, as shown in Table 2. The one-equation transition model in HAM2D was able to capture a reasonable maximum lift-to-drag ratio only at $Re = 3 \times 10^{6}$; the prediction becomes progressively worse compared to experimental data and all other simulation results at higher Reynolds numbers. On the other hand, the two-equation transition model in HAM2D shows fairly good agreement compared to both experiment and other simulation results upto $Re = 9 \times 10^{6}$. The prediction of the linear slope and the maximum $L/D$ value are comparable with those of the $e^{N}$-based transition models from Pires et al. (2016). At $Re = 12 \times 10^{6}$ and $Re = 15 \times 10^{6}$, the two-equation model in HAM2D predicts a lower linear slope than all the reference results and the angle of attack for the maximum $L/D$ is delayed. However, the results from two-equation model are much more in agreement with all reference data in Fig. 6 than the one-equation model over the entire range of Reynolds numbers.

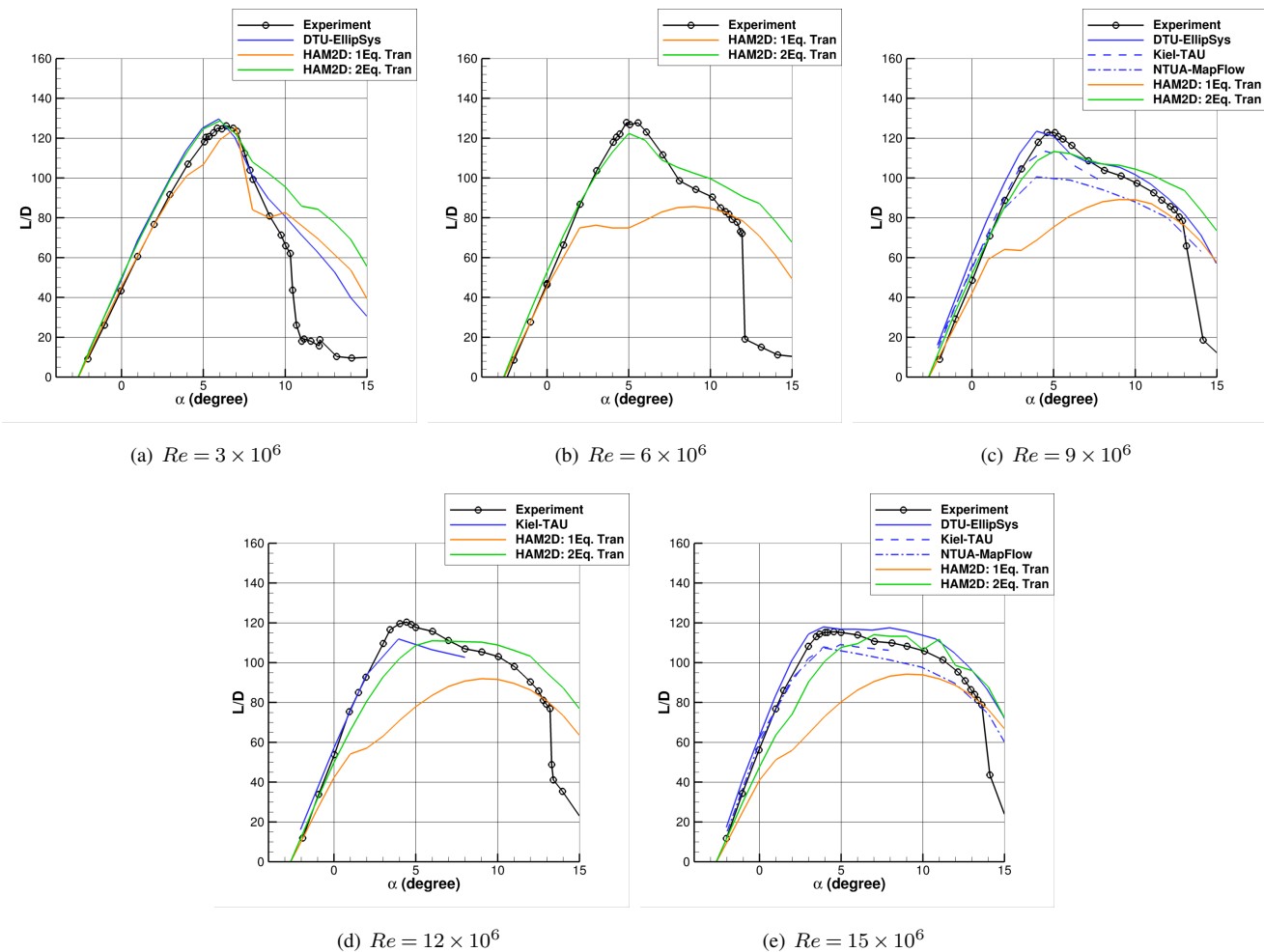

(a) $Re = 3 \times 10^6$

(b) $Re = 6 \times 10^6$

(c) $Re = 9 \times 10^6$

(d) $Re = 12 \times 10^6$

(e) $Re = 15 \times 10^6$

**Figure 6.** Comparison of the lift-to-drag ratio predicted by HAM2D using the one-equation and the two-equation transition models for the DU00-W-212 airfoil against experimental data from Ceyhan et al. (2017a). Simulation results using various transition models from DTU, Kiel, and NTUA (Ceyhan et al., 2017b) are also shown. All simulations are performed at a free-stream turbulent intensity corresponding to Ti3 from Table 2.

To find the reason for underprediction of the lift-to-drag ratio using both transition models, we compare the drag polars from HAM2D predictions and the reference results in Fig. 7. At $Re = 3 \times 10^6$, the HAM2D results using both transition models predict the laminar drag bucket well. As the Reynolds number increases above $3 \times 10^6$, the one-equation model consistently overpredicts the minimum drag while the range of angle of attack of the drag bucket becomes much smaller than the reference

data. This explains the significant underprediction in lift-to-drag ratio at higher Reynolds numbers by the one-equation model. On the other hand, the two-equation model reasonably predicts the experimental drag values upto $Re = 9 \times 10^6$ compared to

the other simulation results. At $Re = 12 \times 10^6$, the minimum drag is overpredicted by 6 drag counts, and the sharp corner of the laminar bucket is not properly captured. More deviation is observed at $Re = 15 \times 10^6$.

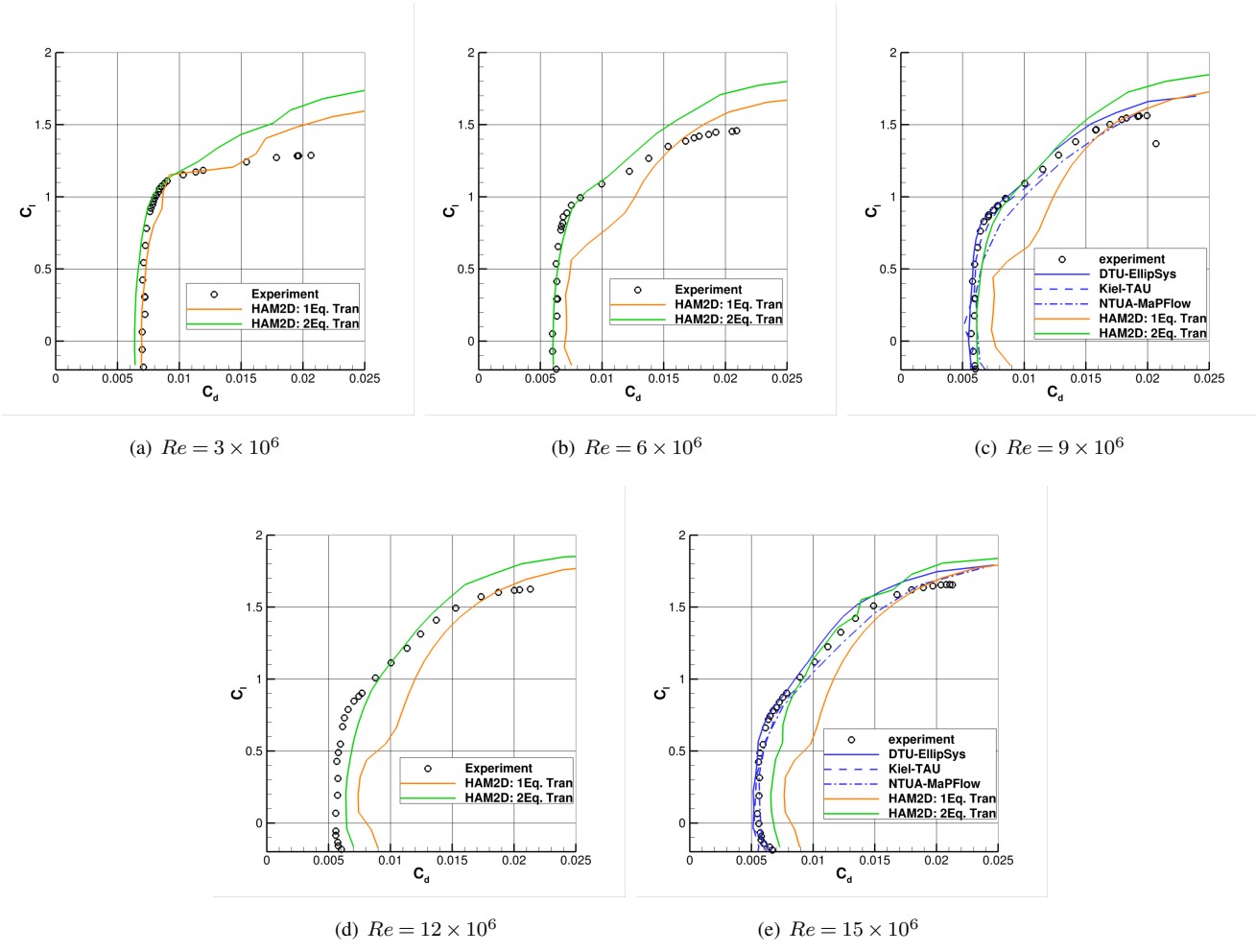

(a) $Re = 3 \times 10^6$

(b) $Re = 6 \times 10^6$

(c) $Re = 9 \times 10^6$

(d) $Re = 12 \times 10^6$

(e) $Re = 15 \times 10^6$

**Figure 7.** Comparison of drag polar predicted by HAM2D using the one-equation and the two-equation transition models for the DU00-W-212 airfoil against experimental data from Ceyhan et al. (2017a). Simulation results using various transition models from DTU, Kiel, and NTUA (Ceyhan et al., 2017b) are also shown. All simulations are performed at a free-stream turbulent intensity corresponding to Ti3 from Table 2.

For the DU00-W-212 airfoil, the drag coefficients at varying Reynolds number are compared with experiment at $4°$ angle of attack where the maximum L/D ratio occurs as shown in Fig 8. It is shown that the drag coefficient from the experiment decreases from 3 to 9 million Reynolds numbers, and then it increases until 15 million Reynolds number. However, the variations between the Reynolds numbers are minor. For the two-equation model, the variations between the Reynolds numbers are also minor as experiment though the drag increases as Reynolds number increases as shown in Fig. 8 (a). Otherwise, the drag

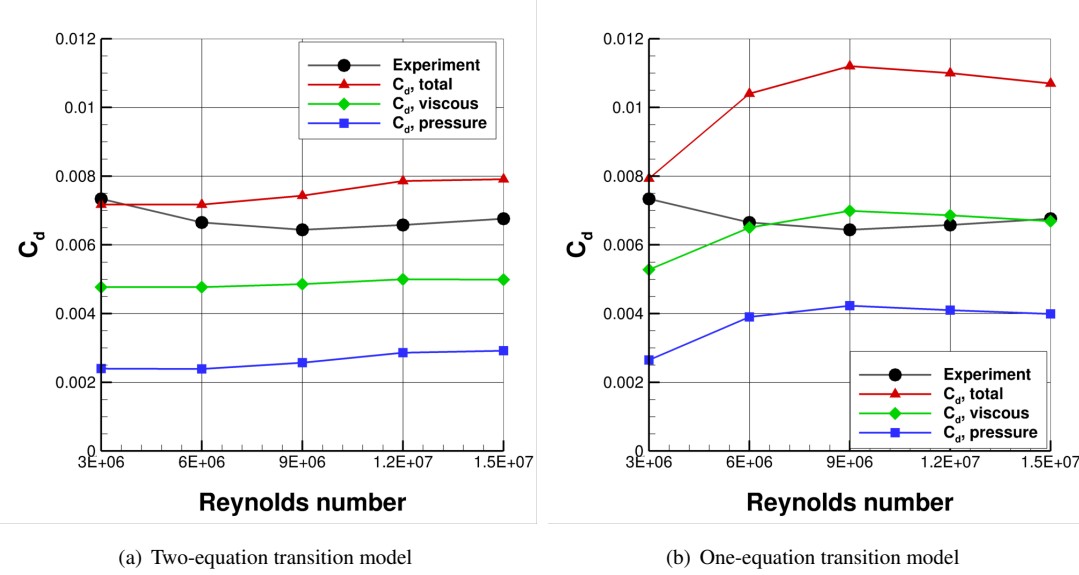

(a) Two-equation transition model      (b) One-equation transition model

**Figure 8.** Drag coefficient breakdown for DU00-W-212 airfoil and comparison with experiment at $4°$ angle of attack and various Reynolds number

clearly increases as Reynolds number increases in the one-equation model prediction, which is opposite trend with experiment as shown in in Fig. 8 (b). Also, the predicted drags are broken down into viscous and pressure drag components. As a result, the viscous drag component is dominant over the pressure drag at all Reynolds numbers from both transition models. This also indicates the importance of transition onset predictions because the skin friction is much higher in a turbulent than laminar boundary layer.

Figure 9 compares the transition onset location, $X_T$, predicted by HAM2D using both transition models on upper and lower sides of the DU00-W-212 airfoil at two representative Reynolds numbers: $3 \times 10^6$ and $9 \times 10^6$. These predictions are also compared with those from EllipSys2D using the $k$-$\omega$-SST turbulence model and different transition models: $\gamma - \overline{Re_{\theta t}}$ (LCTM), $e^N$ model, and the $e^N$-BP model with bypass transition (Sorensen et al., 2014).

At $Re = 3 \times 10^6$, the predicted transition onset locations from HAM2D using both transition models compare well with results from EllipSys2D, as shown in Fig. 9 (a) and (b). As the angle of attack increases, the onset location moves due to the changes in the streamwise pressure gradient similar to the behavior in the S809 airfoil in Fig. 4.

However, at $Re = 9 \times 10^6$, larger deviations start to occur between the predictions from the one-equation and two-equation model on both upper and lower surfaces, as shown in Fig. 9 (c) and (d). Using the one-equation model, the onset prediction rapidly moves to the stagnation point at the $2°$ angle of attack on the upper surface while showing erratic behavior on the lower surface. This explains the early escape of the laminar drag bucket and significant overprediction in drag by the one-equation model compared to the experimental data. Similarly, the larger deviations are also observed among the EllipSys predictions (Sorensen et al., 2014) at $Re = 9 \times 10^6$. The $\gamma - \overline{Re_{\theta t}}$ (LCTM) predicts the onset earlier than $e^N$ and $e^N$-BP models on both

upper and lower surfaces. It is also observed that the bypass transition starts playing a role over the natural transition at this higher Reynolds number by showing the earlier onset prediction using the $e^N$-BP model instead of the $e^N$ model. The two-equation model predictions show a consistent trend in the movement of the onset location, and the results are quite similar to the results from the $e^N$-BP transition model on both surfaces.

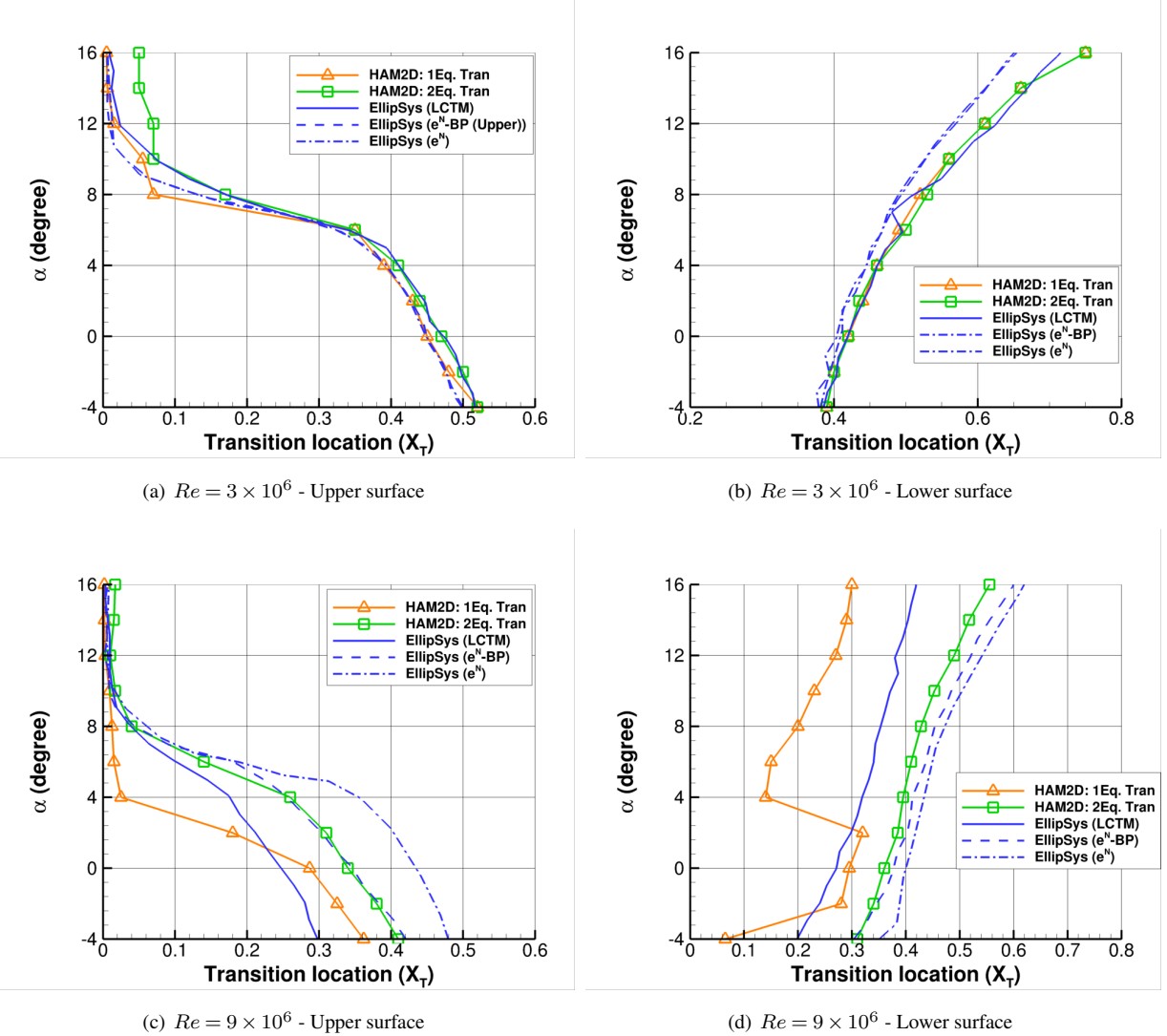

(a) $Re = 3 \times 10^6$ - Upper surface

(b) $Re = 3 \times 10^6$ - Lower surface

(c) $Re = 9 \times 10^6$ - Upper surface

(d) $Re = 9 \times 10^6$ - Lower surface

**Figure 9.** Comparison of variation of transition onset location with angle of attack for the DU-00-W212 airfoil predicted by HAM2D using one-equation and two-equation transition model with that by EllipSys2D using different transition models (Sorensen et al., 2014): $\gamma - \overline{Re_{\theta t}}$ (LCTM), $e^N$ model, and $e^N$-BP model with bypass transition.

Finally, the effect of different inflow turbulence intensity on the predictions from the two-equation transition model is shown in Fig. 10 at two different Reynolds numbers of $3 \times 10^6$ and $12 \times 10^6$. The predicted airloads are compared from the three

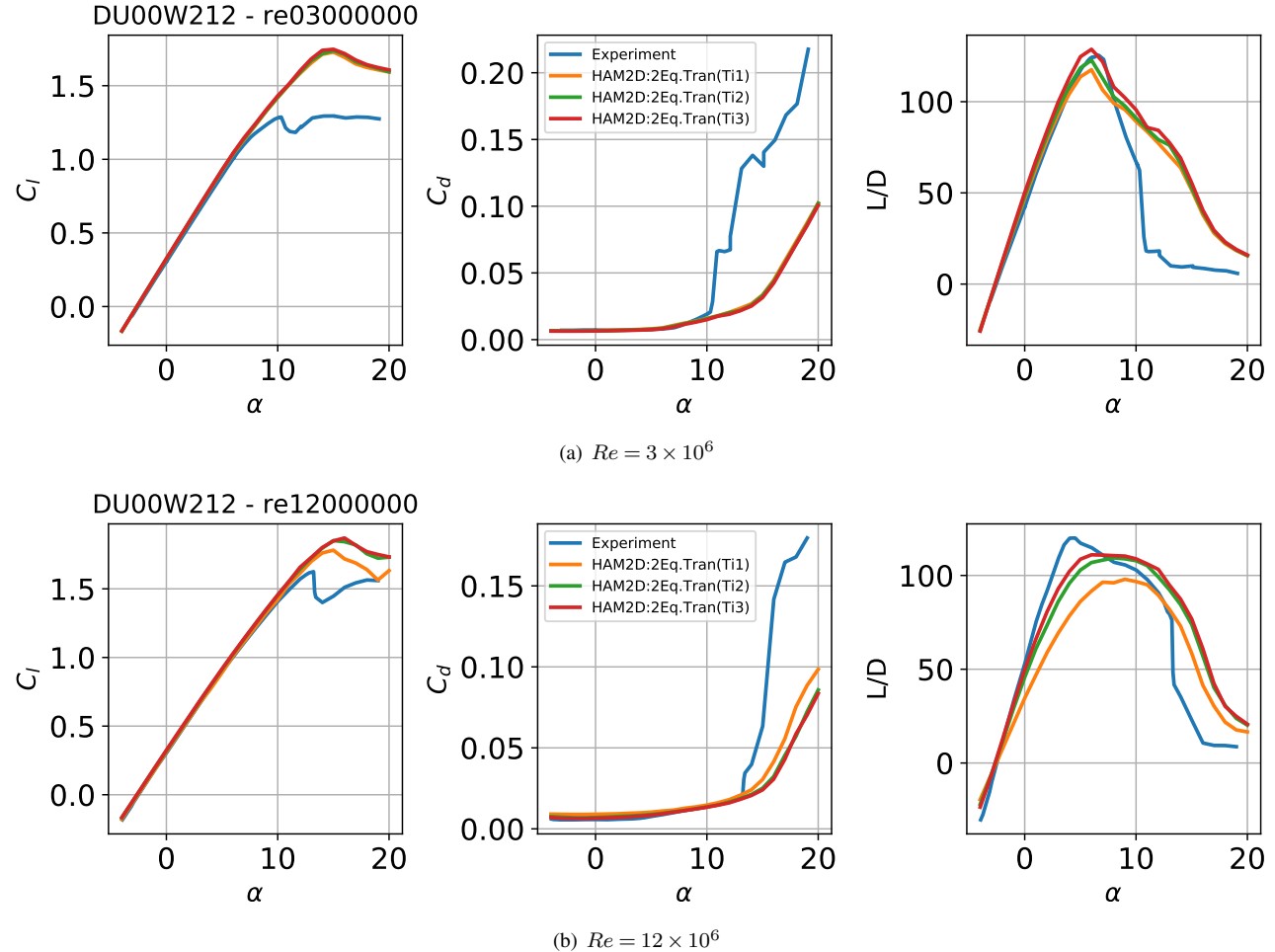

(a) $Re = 3 \times 10^6$

(b) $Re = 12 \times 10^6$

**Figure 10.** Sensitivity of the predictions from HAM2D using the two-equation transition model to the free-stream turbulence intensity (Ti1, Ti2, Ti3) at two Reynolds numbers $Re = 3, 12 \times 10^6$, as defined in Table 2.

different turbulent intensity levels from Ti1 to Ti3, as shown in Table 2. The prediction of the lift-to-drag ratio is highly sensitive to the inflow turbulent intensity level. Also, the sensitivity becomes stronger at the higher Reynolds number, which results in the best correlation with the experiment using the lowest intensity (Ti3) as observed in a previous study using the $e^N$ transition model (Ceyhan et al., 2017b).

### 4.3 DU series airfoils and NACA64-618

The predictions of HAM2D using both transition models for the airfoils in the NREL 5 MW turbine (Jonkman et al., 2009) are compared against data available in the DOWEC 6MW pre-design report (Kooijman et al., 2003) for Reynolds numbers of 6 and 7 million. The reference data for the DU airfoils at $Re = 3 \times 10^6$ are taken from experiments in the LTT wind tunnel of TU Delft. The results for the $Re = 7 \times 10^6$ are the result of a synthesis process, in which measured data for at $Re = 3 \times 10^6$ are

translated to the higher Reynolds number using the airfoil design code RFOIL (Van Rooij, 1996). According to the DOWEC 6MW pre-design report, the reference data for the NACA64-618 airfoil is obtained from appendix IV of Abbott and von Doenhoff (1959). Also, the available reference data was corrected for a blade aspect ratio of 17 in the DOWEC 6MW pre-design report (Kooijman et al., 2003). However, we believe the data is still valid as a reference in explaining any differences of model predictions.

The automated grid generation for these airfoils uses 400 points in the wrap-around direction based on the grid-refinement study shown in Appendix B. The free-stream turbulence intensity is set to 0.1%. Figures 11 show the comparison of fully turbulent and free-transition results for DU91-W2-250 airfoil at $Re = 7 \times 10^6$ and NACA64-618 airfoil at $Re = 6 \times 10^6$ against reference data from Kooijman et al. (2003).

All simulation results, both using the fully turbulent and transition models, show similar behavior and predict the lift coefficient well in the linear region including the lift-curve slope and the zero-lift angle of attack. Overall, the current simulations predicted delayed stall angles compared to the reference data. It should be noted that the same trend is also observed in the previous comparison with pure experimental data for the DU00-W-212 airfoil in Fig. 10, which is a typical challenge in 2D RANS-CFD modeling of airfoils. The unphysical linear increment in the drag coefficient is observed after the stall angle only in the reference data. This might be due to the synthesized process between RFOIL calculations and experimental data.

By using either the one- or two-equation transition model, lower drag coefficients were predicted at around $0°$ as a result of laminar boundary layer detection. This results in a better agreement in lift-to-drag ratio against reference data compared to the fully turbulent simulations. The prediction of the maximum lift-to-drag ratio is significantly improved using the two-equation model compared to the one-equation model for both airfoils. The one-equation model underpredicts the lift-to-drag ratio in the linear portion of the lift curve due to early transition onset as the angle of attack increases. Thus, the two-equation transition model is an appropriate choice for wind-turbine airfoil simulations at high Reynolds numbers.

Similar comparison results for the other NREL 5MW airfoils with different maximum relative thickness $(t/c)_{\mathrm{max}}$ are shown in Appendix A1. Overall, two-equation transition model improves the predictions for the other airfoils as well.

## 4.4 FFA series airfoils

We compare the predictions of HAM2D using both transition models for the FFA-W3 series of airfoils in the DTU 10 MW (Bak et al., 2013) and the IEA 15 MW turbine (Gaertner et al., 2020) at a Reynolds number of 10 million. We also compare our results against the publicly available simulation data from EllipSys2D for these airfoils (Gaertner et al., 2020) using the $k$-$\omega$-SST turbulence model with the semi-empirical $e^{\mathrm{N}}$ method (Drela and Giles, 1987). However, only a combination of 70% free-transition/30% fully turbulent polar is available for these airfoils, i.e. the lift and drag values at each angle of attack are linearly interpolated between the free-transition and full-turbulent results using the 70/30 ratio. Therefore, we generate the same mixed polars in this study by using one-equation and two-equation transition model for appropriate data comparison. The automated grid generation for these airfoils uses 400 points in the wrap-around direction based on the grid-refinement study shown in appendix B. The free-stream turbulence intensity is set to 0.1%.

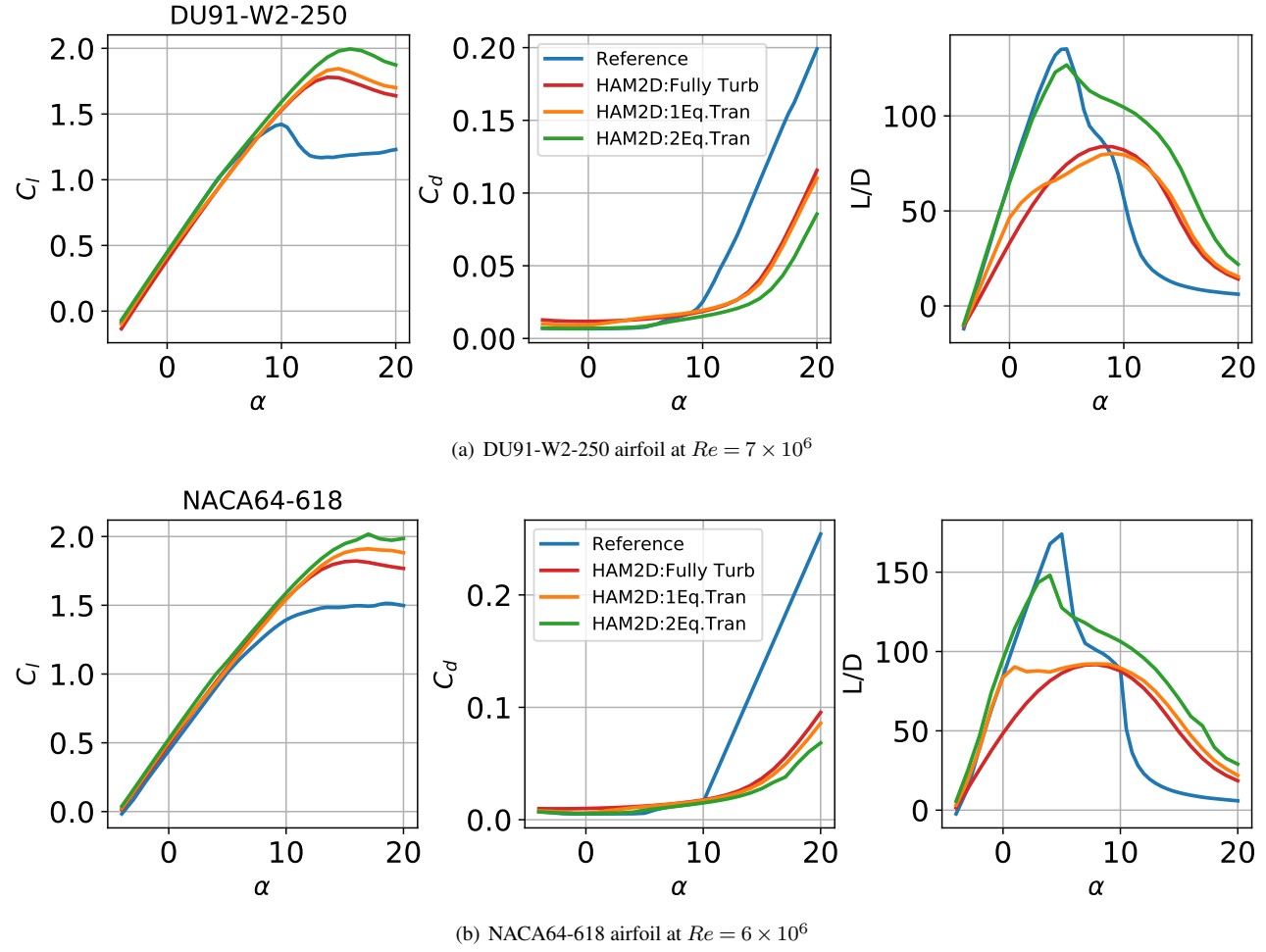

(a) DU91-W2-250 airfoil at $Re = 7 \times 10^6$

(b) NACA64-618 airfoil at $Re = 6 \times 10^6$

**Figure 11.** Aerodynamic coefficient polars for DU91-W2-250 and NACA64-618 airfoils using fully turbulent and free transition at $Re = 7 \times 10^6$ and $6 \times 10^6$ generated using HAM2D compared against reference data from Kooijman et al. (2003).

For the FFA-W3-301 airfoil, our simulation results are compared with simulation results from EllipSys2D (Gaertner et al., 2020) as shown in Fig. 12. The predictions using HAM2D with the two-equation model show excellent agreement with the reference. The one-equation model predicted much lower lift-to-drag ratio than the predictions from other transition models in the linear portion of the lift curve due to earlier transition onset. This is similar to the behavior seen in the DU00-W-212 (Sec. 4.2) and the NREL 5 MW airfoils (Sec. 4.3 and Appendix A1).

Similarly, the comparison results for the other FFA-W3 airfoils with different maximum relative thickness $(t/c)_{\mathrm{max}}$ are shown in Appendix A2. Overall, the predictions only from two-equation model show excellent agreement with the reference for the other airfoils as well.

     For a better understanding of the two-equation model in HAM2D, we compare its behavior to the implementation of $\gamma - \overline{Re_{\theta t}}$ model in EllipSys2D using the $k$-$\omega$-SST turbulence model for the FFA-W3-301 airfoil (Bak et al., 2013) at $Re = 10 \times 10^6$. The

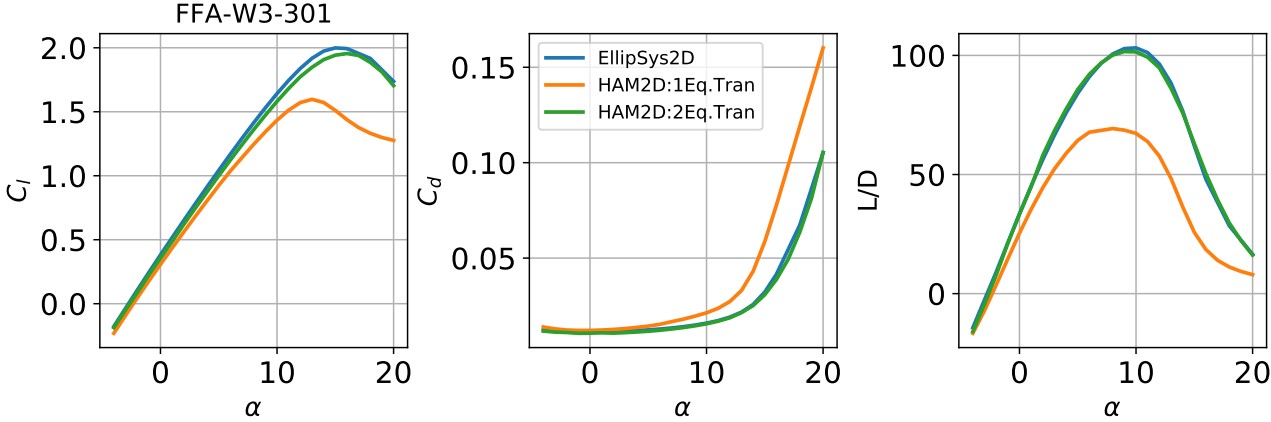

**Figure 12.** Aerodynamic coefficient polars for the FFA-W3-301 airfoil using fully turbulent and free transition at $Re = 10 \times 10^6$ generated using HAM2D compared against a mix of 70%/30% transition/turbulent data from EllipSys2D (Gaertner et al., 2020)

.

skin friction distribution predicted by HAM2D and EllipSys2D for this airfoil are compared at four different angles of attack in Fig. 13. The sign of skin friction is defined by the sign of the local streamwise velocity at each point. The transition onset location is indicated by a sharp increase in the skin friction value on both the upper and lower surfaces. The transition onset prediction from the one-equation model in HAM2D rapidly moves to upstream at the $8°$ angle of attack. As a result, it predicts a delayed onset at the lower angles of attack, but earlier onset at the higher angles of attack compared against the EllipSys2D result. On the other hand, the transition onset predicted by the two-equation model is downstream of the predictions from the one-equation model and from EllipSys2D at all angles of attack. A similar behaviors was also observed for the DU-00-W212 airfoil at $Re = 9 \times 10^6$ in Fig. 9. The delayed onset locations from the two-equation model in HAM2D than other LCTM predictions might explain the good airload agreement with $e^N$ method as shown in Fig. 12.

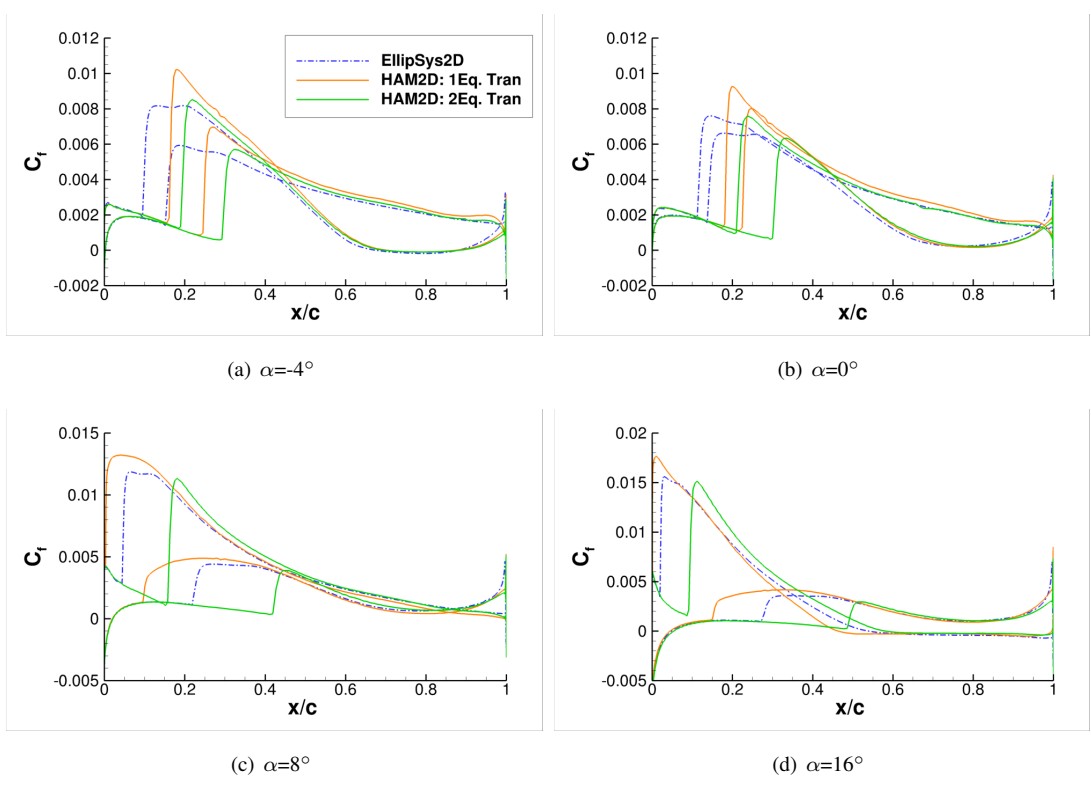

**Figure 13.** Skin friction coefficient distribution for the FFA-W3-301 airfoil at $Re = 10 \times 10^6$ generated using HAM2D with the one-equation and two-equation transition models compared against EllipSys2D with the $\gamma - \overline{Re_{\theta t}}$ transition model (Bak et al., 2013).

## 5    Conclusions

We evaluated the performance of two local correlation-based transition models within our in-house 2D compressible Reynolds-averaged Navier-Stokes (RANS) solver HAM2D for applications to modern wind-turbine airfoils at high Reynolds numbers. The one-equation transition model ($\gamma-$SA) and the two-equation transition model ($\gamma-\overline{Re_{\theta t}}-$SA) are coupled with the Spalart-Allmaras (SA) one-equation turbulence model. We compare the predictions of the two transition models with available experimental and computational fluid dynamics (CFD) data in the literature in the Reynolds number range of 3-15 million including the AVATAR project measurements of the DU00-W-212 airfoil (Ceyhan et al., 2017a) and for airfoils from three modern, open source, MW-scale wind turbines, NREL 5 MW (Jonkman et al., 2009), DTU 10 MW (Bak et al., 2013), and IEA 15 MW (Gaertner et al., 2020).

The two models exhibit similar behavior at Reynolds numbers around 3 million. The one-equation transition model fails to predict the natural transition behavior at the high Reynolds numbers ranging from 6 million to 15 million due to early transition onset, as reported in previous study for $\gamma-\overline{Re_{\theta t}}$ model (Sorensen et al., 2016). The two-equation transition model presents much better predictions in aerodynamic coefficients (e.g. stall angle, maximum lift coefficient, and lift-to-drag ratio) than the one-equation transition model. As a result, comparable performance with the $e^{\text{N}}$-based transition models within RANS-CFD are observed for the various thickness airfoils. At high Reynolds numbers from 12 million, the two-equation model also somewhat underpredicted the maximum lift-to-drag ratio compared to the results from $e^{\text{N}}$-based transition models.

The one-equation transition model also fails to predict the correct trends of the aerodynamic coefficients, especially the peak lift-to-drag ratio, with Reynolds number. On the other hand, the predictions in aerodynamics coefficients at all Reynolds numbers from the two-equation transition model are much closer to that of the experimental data and comparable to the predictions from the $e^{\text{N}}$-based models in the literature (Ceyhan et al., 2017b). The predictions from the two-equation transition model exhibits a strong sensitivity to the free-stream turbulence intensity at the high Reynolds number, as previously observed from the $e^{\text{N}}$-based models. Overall, the combination of the two-equation transition model coupled with the Spalart-Allmaras RANS turbulence model is a good method for performance prediction of modern wind-turbine airfoils using CFD.

The shortcomings of the one-equation transition model at high Reynolds numbers have been identified by comparing against the two-equation transition model. However, the formulation of one-equation transition model satisfies Galilean invariant which is desirable in a simulation with rotating bodies (e.g. blade). Therefore, in the future, we plan to improve the performance of the one-equation transition model using the Field-Inversion Machine-Learning approach which was validated for the SA turbulence model (Holland et al., 2019).

## Appendix A: Additional Results

The current predictions of HAM2D using one- and two-equation transition models are further evaluated for NREL 5MW (Jonkman et al., 2009), DTU 10 MW (Bak et al., 2013), and IEA 15 MW (Gaertner et al., 2020) airfoils which have different maximum relative thickness, $(t/c)_{\max}$.

### A1 DU series airfoils

Figures A1 show the comparison of fully turbulent and free-transition results for DU airfoils at $Re = 7 \times 10^6$ against reference data from Kooijman et al. (2003). Overall, much better agreements in lift-to-drag ratio against experimental data are observed by using the two-equation transition model for all airfoils. Also, the lift curve slopes in the linear region are better matched with the experiment using two-equation transition model for airfoils with larger maximum relative thickness $(t/c)_{\max}$.

The difference between the predictions of the maximum $L/D$ and lift curve slope from the one-equation and two-equation models increases for larger thickness airfoils. This might be because the onset location typically moves towards the leading edge for the thicker airfoils due to the higher adverse pressure gradient at the same angle of attack. A more detailed discussion can be found in Section 4.3.

### A2 FFA series airfoils

For the FFA-W3 airfoils with different airfoil thickness, our simulation results using both transition models are compared with simulation results from EllipSys2D (Gaertner et al., 2020) using the semi-empirical $e^{\mathrm{N}}$ method, as shown in Fig. A2. In this comparison, a combination of 70% free-transition and 30% fully turbulent polars are used as we already discussed in the Section 4.4. For all FFA series airfoils, the predictions using HAM2D with the two-equation model show excellent agreement with the reference except in the case of the very thick FFA-W3-360 airfoil. Like the comparison in the Section 4.4 for FFA-W3-301 airfoil, the one-equation model predicted much lower lift-to-drag ratio than the predictions from two-equation transition model for all airfoils. A more detailed discussion can be found in Section 4.4.

## Appendix B: Grid Convergence Study

A grid convergence study was conducted to measure the sensitivity of airfoil performance to grid resolution to validate the current grid resolution. The grid convergence study was performed for the airfoils using different number of surface points: 300, 400, 500, and 600. The initial wall normal spacing was fixed with a small enough value such that $y^+ = 1$. The test was focused at a specific operating flow condition with $\alpha = 4°$ and $Re = 9 \times 10^6$. The simulations are performed for both the fully turbulent and free-transition boundary layer. The results for the DU93-W-210LM airfoil are shown in Figs. B1, B2, B3. The y-axis limit is the ranges of $\pm 1\%$ of each mean changes in the number of grid points. It is seen that the magnitude of the variation is less than 1% of their actual values, which results in minor variation compared to the variation from different airfoils or flow conditions of the current interest.

## Appendix C:  Solution Convergence Study

Figures C1 and  C2 show the solution convergence history during the simulation for the representative flow condition at two different Reynolds numbers of $3\times10^6$ and $9\times10^6$. Both lift and drag coefficients are converged within 1500 iterations, wherein the solution residual drops by more than 3 orders of magnitude.

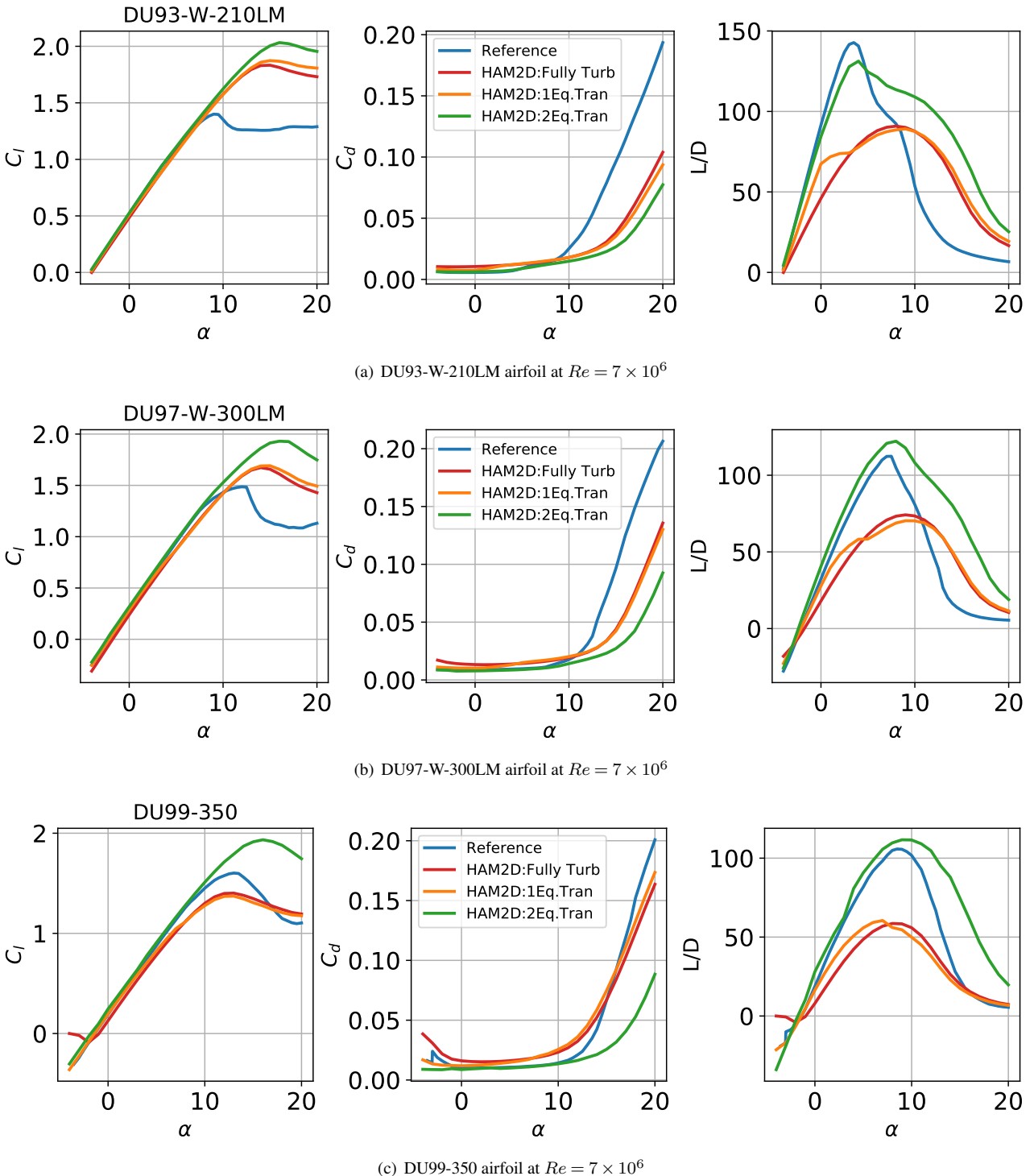

(a) DU93-W-210LM airfoil at $Re = 7 \times 10^6$

(b) DU97-W-300LM airfoil at $Re = 7 \times 10^6$

(c) DU99-350 airfoil at $Re = 7 \times 10^6$

**Figure A1.** Aerodynamic coefficient polars for DU airfoil series using fully turbulent and free transition at $Re = 7 \times 10^6$ generated using HAM2D compared against reference data from Kooijman et al. (2003).

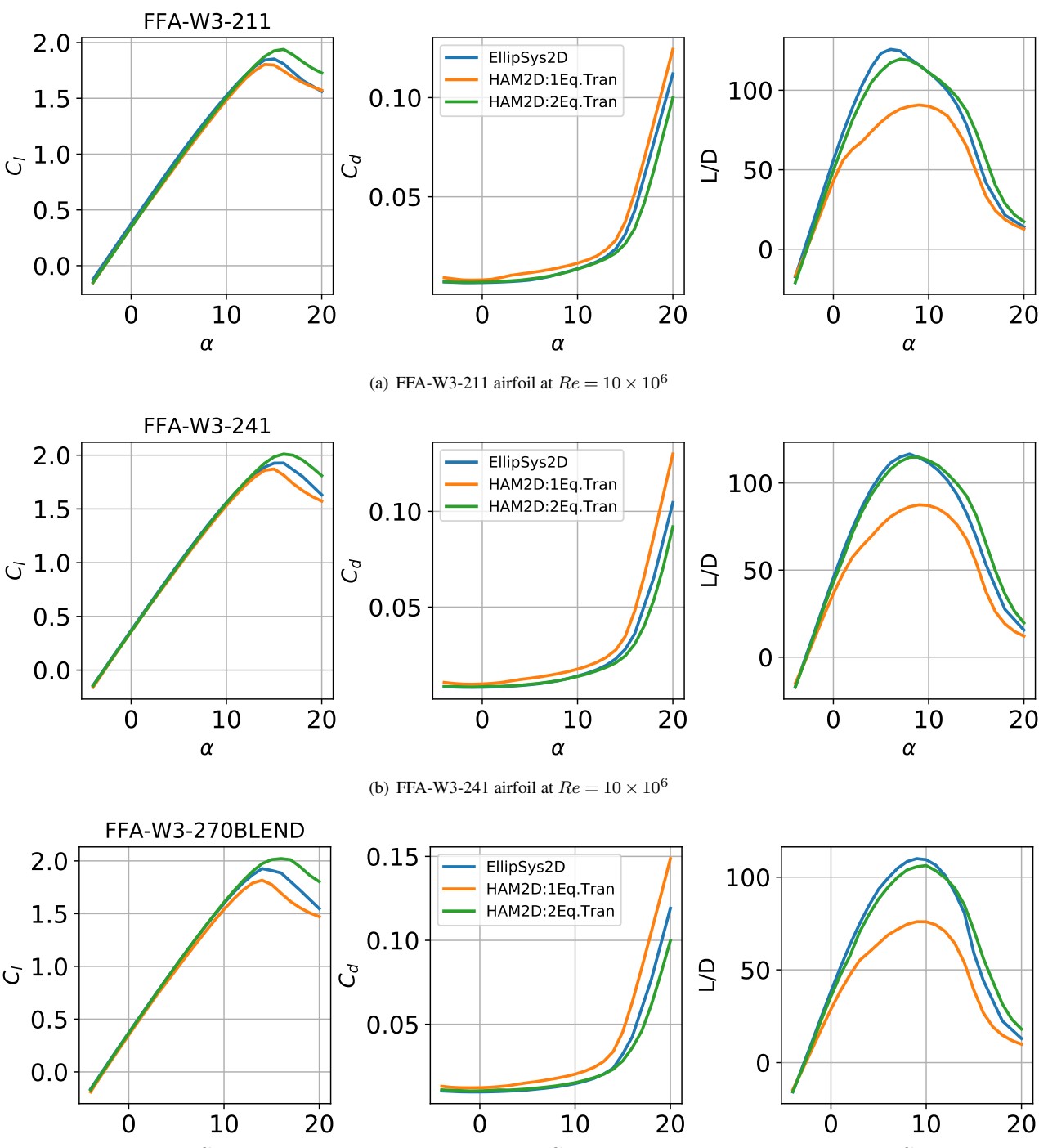

(a) FFA-W3-211 airfoil at $Re = 10 \times 10^6$

(b) FFA-W3-241 airfoil at $Re = 10 \times 10^6$

(c) FFA-W3-270BLEND airfoil at $Re = 10 \times 10^6$

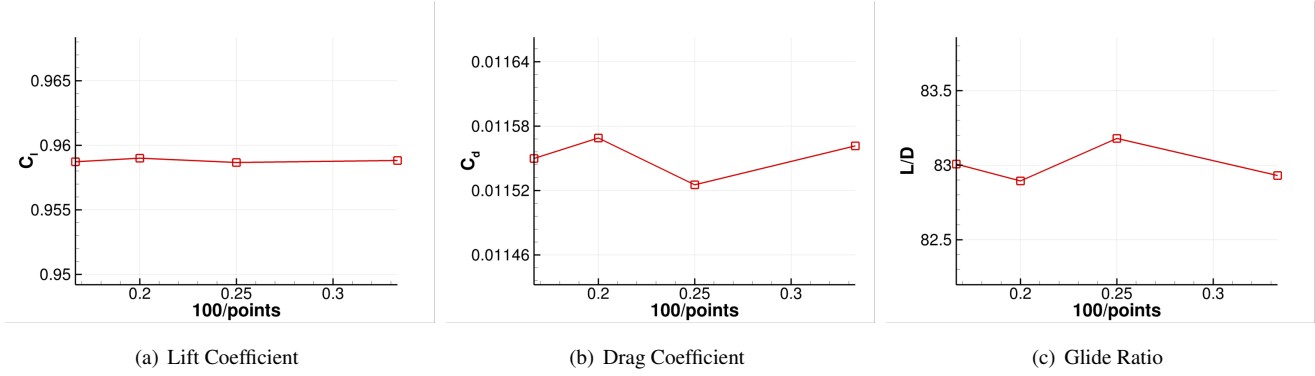

(d) FFA-W3-330BLEND airfoil at $Re = 10 \times 10^6$

(e) FFA-W3-360 airfoil at $Re = 10 \times 10^6$

**Figure A2.** Aerodynamic coefficient polars for the FFA airfoil series using fully turbulent and free transition at $Re = 10 \times 10^6$ generated using HAM2D compared against a mix of 70%/30% transition/turbulent data from EllipSys2D (Gaertner et al., 2020).

(a) Lift Coefficient

(b) Drag Coefficient

(c) Glide Ratio

**Figure B1.** Grid resolution study for the DU93-W-210LM airfoil at $\alpha=4°$ and $Re$=9M (fully turbulent flow)

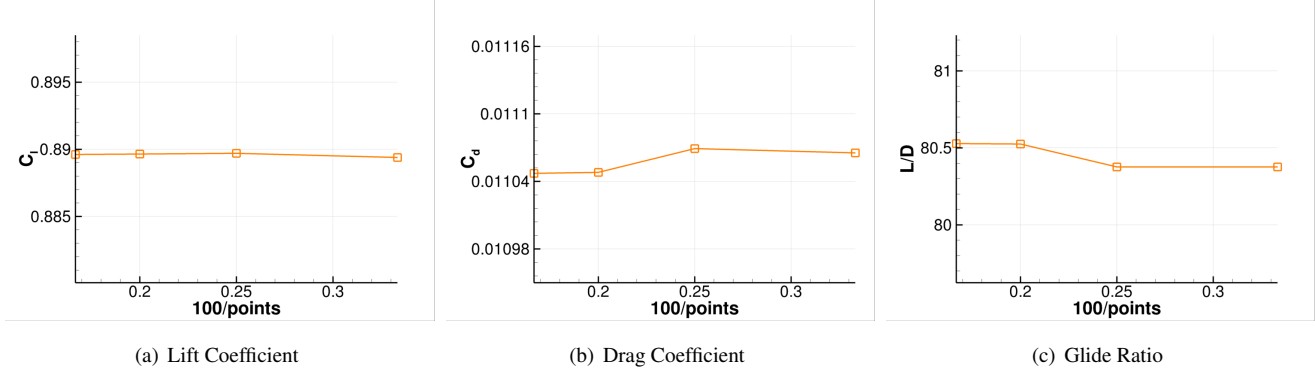

(a) Lift Coefficient        (b) Drag Coefficient        (c) Glide Ratio

**Figure B2.** Grid resolution study for the DU93-W-210LM airfoil at $\alpha$=4° and $Re$=9M (free-transition using the one-equation model)

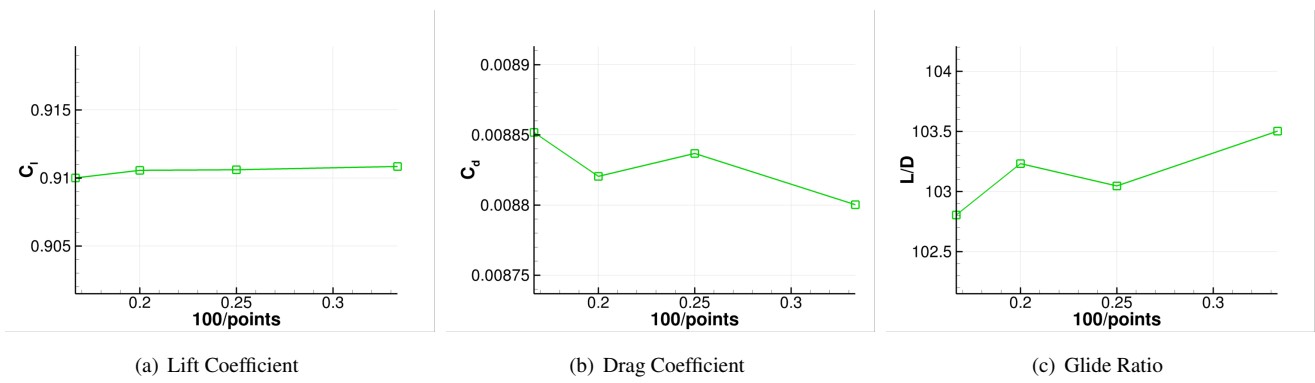

(a) Lift Coefficient        (b) Drag Coefficient        (c) Glide Ratio

**Figure B3.** Grid resolution study for the DU93-W-210LM airfoil at $\alpha$=4° and $Re$=9M (free-transition using the two-equation model)

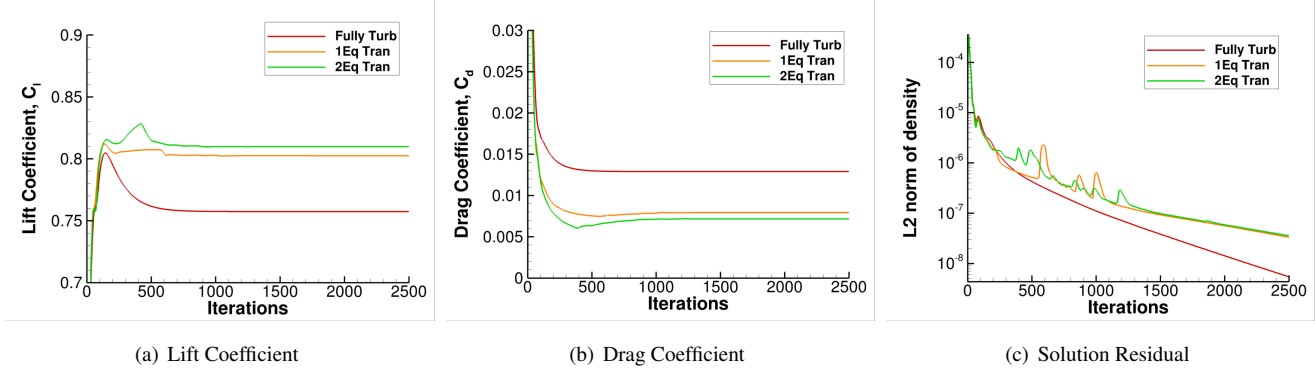

(a) Lift Coefficient        (b) Drag Coefficient        (c) Solution Residual

**Figure C1.** Solution convergence history during simulations for the DU-00-W212 airfoil at $\alpha$=4° and $Re$=3M

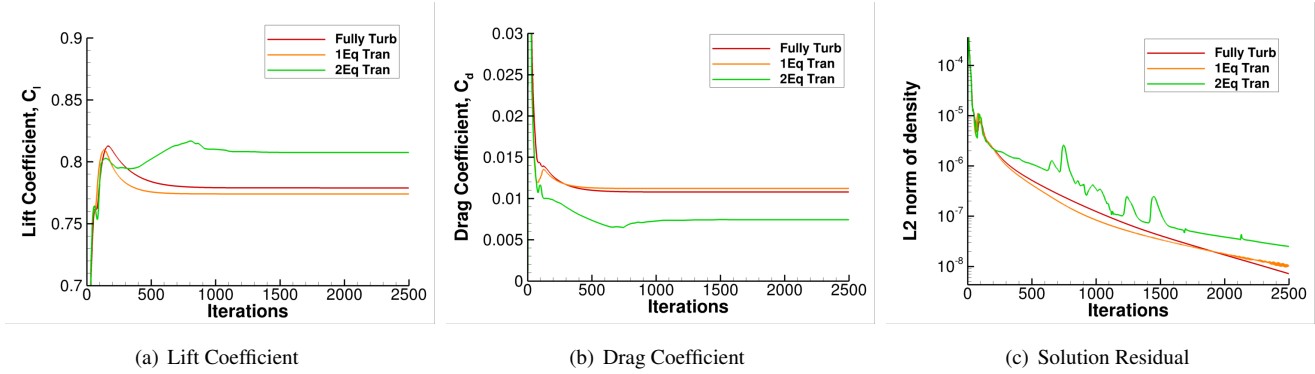

(a) Lift Coefficient          (b) Drag Coefficient          (c) Solution Residual

**Figure C2.** Solution convergence history during simulations for the DU-00-W212 airfoil at $\alpha$=4° and $Re$=9M

*Data availability.* Data for any plots in this manuscript can be made available on request

*Author contributions.* YSJ developed the simulation code and performed the simulations. YSJ wrote the manuscript with significant input from GV. GV provided reference data set and developed the post-processing code for the simulation results. SA and JB provided guidance for the research and reviewed the manuscript.

*Competing interests.* The authors declare that they have no conflict of interest.

*Acknowledgements.* This work was authored in part by the National Renewable Energy Laboratory, operated by Alliance for Sustainable Energy, LLC, for the U.S. Department of Energy (DOE) under Contract No. DE-AC36-08GO28308. Funding provided by the Advanced Research Projects Agency-Energy (ARPA-E) Design Intelligence Fostering Formidable Energy Reduction and Enabling Novel Totally Impactful Advanced Technology Enhancements (DIFFERENTIATE) program. The views expressed in the article do not necessarily represent
the views of the DOE or the U.S. Government. The U.S. Government retains and the publisher, by accepting the article for publication, acknowledges that the U.S. Government retains a nonexclusive, paid-up, irrevocable, worldwide license to publish or reproduce the published form of this work, or allow others to do so, for U.S. Government purposes. A portion of this research was performed using computational resources sponsored by the Department of Energy's Office of Energy Efficiency and Renewable Energy and located at the National Renewable Energy Laboratory. The transition modeling work was supported under the Vertical Lift Research Center of Excellence Grant at the
University of Maryland with Dr. Mahendra Bhagwat as Technical Mentor. This work was supported by Pusan National University Research Grant, 2021.

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
