# Peer review of "Local Correlation-based Transition Models for High-Reynolds-Number Wind Turbine Airfoils"

_Wind Energy Science, 2021_

## Referee Comment (RC1)

**Review of WES-2021-23 manuscript**

**General Overview**

WES manuscript 2021-23 provides potentially interesting research results of the usage of two equation correlation based laminar-turbulent transition models implemented in Spalart Almaras turbulence model for the wind turbine airfoils at high Reynolds numbers. Authors compare the results both with the experimental data of a few airfoils and with the other CFD results in the literature. These comparisons can provide good insights and possibilities on the usage of two equation transition models for the predictions of the high Reynolds number flows around large offshore wind turbine airfoils. However, some of the experimental data used are questionable, some sections are not well structured, additional literature beyond the wind energy applications are necessary for the transition models and there are also many points that need clarification and correction. All these are needed to be completed and clarified before this manuscript can be considered.

Recommendation: Major revision

**Specific Comments**

Line 33-34: I am not sure where in the paper of Sorensen et all 2016 this is stated. The main goal of that study was to make sure the codes produce consistent solutions in terms of domain size, grid resolution, convergence criteria etc. Moreover, there were no experimental results to compare. Another look at the literature is required to support this statement.

Line 34-35: accurate prediction of the glide ratio near design points.

Line 35: what/where is the design operating point? Normally, the airfoils are designed for a range of angle of attacks not for a single point.

Line 35: Please check cross-reference formats throughout the paper. Is it Sorensen (2014) or (Sorensen 2014)? I won't point out all of them but please check and correct the whole paper for this.

Line 35-36-37: In this paper of Sorensen, there were only three codes with transition model and all of them used eN method. Two of them provided results with different grid resolutions. Although it can be seen there is an effect of the eN transition prediction in the glide ratio and trends with Re numbers, I don't think this is a sufficient evidence for this statement; especially since in that paper there is no experimental results to compare. How about the other transition models used in the literature? For example, as you have also pointed out, Sorensen 2014 showed that the correlation based transition model of Menter was giving wrong trends wrt increasing Re number.
There is also another stability theory based transition model used in the paper from Coder. How about this one? It is recommended to check these statements in this paragraph and support with more evidence.

Line 40: "is difficult" - why it is difficult? Do you mean it takes longer to converge? Or something else? Please clarify

Line 41-42: what is wrong with computing N from semi-empirical models or from a stability solver? Please clarify.

Line 44: ... in coupling it with ...

Line 45-46: How about the suitability of this method? What is right/wrong with it? Was it used for the wind turbine airfoils applications?

Line 74: how about the suitability of the method for the machine learning process as mentioned in the beginning as the goal of the study?

Line 125 - paragraph. This paragraph seems incomplete. what is ysep and Gonset? Why these are not needed? What is the advantage or disadvantage?

General question about the implementation of the two eqn. transition model: was there a need to modify the original correlations or a new calibration to obtain better results for high Re number cases? Should we expect any influence?

Line 164: Why start with "However"?  If the turbulence intensity is not a variable in the SA turbulence model and the current study is also assumes that the measured turbulence intensity is constant everywhere, this should be convenient for the SA turbulence model right?

Line 170: Validation. This section is hard to follow. A few subsections would help the reader to find its way easier.

Line 189-190: "... This implementation of the SA model..." this sentence seem redundant. We already know this.

Line 183: What can you say about the comparisons about the grid resolution and other parameters used?

Figure 3: Legend should be HAM2D and not Fully turb

Line 191: in this paragraph it is not clear whether you run Overflow yourself or you used the data from the transition modelling workshop.

Line 194: What kind of transition model is this AFT2019?

Line 195: "... Test flow condition is at free stream... " I don't understand this sentence. What is two equation Mach number?

Line 202: Do you also use the simulation results from Coder 2019 paper? As stated above it would be good to clarify which simulation comes from where and be consistent with how you refer to it in different places in the paper.

Line 204: What are the "known limitations if 2D CFD RANS"? How about whether or not these codes were able to resolve the laminar separation bubble at these simulations? Experimental results from TU Delft Wind tunnel as used by Somers https://www.nrel.gov/docs/legosti/old/6918.pdf show this bubble

clearly and how lift actually continues to rise once post stall after the bubble is gone. I would expect a more elaboration on this physical phenomenon and how these codes were/were not able to capture this.

Line 206: Again in this paragraph, please make sure you refer to the correct references and address where the simulations come from consistently.

Figure 4a: Is the experimental results are for the tripped case? I think both tripped and untripped Cl should be available. Although the difference was small in the case of TU Delft experiments.
How about the Cl results for the negative angle of attack values?

Figure 4 legend of Overflow. It used SA turbulence model so there is no need to write SA again in the legend. Instead best to keep the names of the transition models only and add "fully turbulent" when transition is off.

Line 227: it says you will first study the effect of freestream turbulence intensity but these are not the first results.

Line 271: Ti3 is the lowest turbulence intensity given in the experiment in Table 2. Do you mean you need to lower this also?

Line 272: Laminar drag bucket is not visible in Fig.9.

Figure 6: Pires et all. 2016 does not have any simulation results??

Figure 8: are these still DU212 airfoil? Where do these results come from?

Figure 9 and Line 292: How can you tell that the L/D prediction is highly sensitive to the transition onset locatio; where do you see it in Figure 9?

Line 293: Which both quantitates?

Line 298: TU Delft wind tunnel cannot reach 6 or 7 million Re numbers for airfoil polars. It can go max. Re number of 3.3 million. You should check your reference.

Line 306: "By using either the one- or two-equation
transition model, lower drag coefficients were predicted at around 0 as a result of laminar boundary layer detection. This results in a better agreement in lift-to-drag ratio against experimental data compared to the fully turbulent simulations." These sentences are repetition. You have already emphasized it in the validation section when transition model is used, you have better drag prediction.

Line 311: why is the difference between two transition models increasing going to thicker airfoils? Could you elaborate this?

Figure 10: The experimental data cannot come from TU Delft wind tunnel. You should check your reference.

Line 318: Do you mean for 70% of AoA, the results are from transitional and 30% of Aoa, they are from fully turbulent results?

Line 326: how can you say that one equation model underpredicts the L/D from Fig 11? there is no experimental data...

Line 330: I thought the name of the transition model in HAM2D was Gamma-ReThetat-SA? Are you using also the original Gamma-ReTheta formulation then?

Typo in "EllipSys2D" overall.

Line 340: How can you tell that two equation model predicts transition more accurately by looking at the Fig.11 and Fig.12? There are no experimental results there. And in Fig12, EllipSys results are also with two equation transition model.

General remark on the results section: Since the figures are far from the text, it is difficult to follow it. I also have the impression that not all figures are used in the text; especially of Fig. 11. Perhaps a synthesis can be made for the ease of the reader. Another remark is the legends. Please add HAM2D in the beginning so that it is clear which results we should be looking at.

Line 346: another confusion in the name of the 2eqn. transition model.

Line 349: where does it say these turbines are commercially relevant? Is there a reference for that?

Line 354: They used 2eqn. model in this reference, not 1.

Starting with Line 356: How about the trend with increasing Re number? In the reference Sorensen 2016, they show that two equation model shows increasing drag with increasing Re number. How about in your results? Although you state it is better but from which Figure I can see this?

Line 361: In the reference Ceyhan 2017b, there are no two equation model results. I think you refer to the turbulence intensity values. If so, please rephrase this sentence.

Line 362: Why do you say there is a limitation on the two equation model? How does being sensitive to the turbulence intensity make the model limited?

Line 368: why Galilean invariant formulation makes a model more desirable? From what I can see in these results, there is no reason to use 1eqn. model for these applications.

Line 456: Yilmaz 2017 and Ceyhan 2017b are the same references?

---

## Referee Comment (RC2)

Review of paper WES-2021-23:

Local Correlation-based Transition Models for High-Reynolds-Number Wind Turbine Airfoils

Yong Su Jung[1], Ganesh Vijayakumar[2], Shreyas Ananthan[2], and James Baeder[1]
[1]University of Maryland, College Park, MD
[2]National Renewable Energy Laboratory, Golden, CO

**General remarks.**

The authors compare results from the Spalart-Allmaras RANS turbulence model coupled with a one-equation and two-equation transition model with other CFD codes and experiments for wind turbine airfoil characteristics at high Reynolds numbers. This may give valuable results since the trend in turbine design is towards increasingly higher blade chord based Reynolds numbers, at present already as high as 15 million. Quite some efforts has been spent on computations for various configurations. However, the manuscript suffers from a number of inconsistencies in the use of experimental data. Also the manuscript structure and elaboration on /analysis of differences between results of different codes may be improved. The following will concentrate mainly on the use of the experimental data.

**Details**

The chapter names Validation and Results seem a bit strange without a proper clarification in the name. What is being validated?  Isn't the comparison under the chapter Results with all the comparisons with different airfoil computations and experiments not also a validation?

The paper shows quite some comparisons of calculations, while the reason for the differences sometimes remains unclear. In the paper not a single pressure distribution is shown, while this also may shed some light on the source of the differences. I would opt for less examples and a more thorough investigation of the ins-and outs of the calculations.

**Figure 2 and 10 and chapter 4.2**

There is no such airfoil as DU21-A17, nor do any of the other DU-airfoils mentioned in chapter 4.2 exist. I'm afraid the authors confuse the aero-data file names with the airfoil data of the NREL 5MW turbine with the actual airfoil names. DU21-A17 is actually the file with airfoil characteristics of airfoil DU 93-W-210LM for a blade aspect ratio of 17. The addition  LM stands for a small reduction in the trailing edge thickness done in the framework of the Dutch DOWEC study [1], where this modification originates from. NACA 64-A17 is simply the file with data for the original NACA $64_3$-618 corrected for aspect ratio. The lift, drag and moment coefficients given in the files are synthesized data on the basis of calculations and experiments (at lower Reynolds numbers). They are not the direct result of measurements in a wind tunnel for a Reynolds number of $7\times10^6$. The differences between experiment and computations in figure 10e, however, will still be big, as the maximum lift coefficient for the NACA 64-618 at Re=$6\times10^6$ is only slightly higher than 1.5
Instead maybe some older NACA airfoils can be used, for which characteristics up to Re=$9\times10^6$ are available. If specifically wind turbine airfoils are needed, ref. 2 may be of help.

**Figure 9:**
The authors might wish to check the post-stall drag data of DU 00 the data set. The wake rake had a fixed span position, so in stall that might give values either too high or too low because of 3D stall patterns. Using the pressure drag post stall instead of wake rake data may give at least less dramatic differences.

Recommendation: major revision

[1] Kooijman, H.J.T., Lindenburg, C.,  Winkelaar, D., van der Hooft, E.I.  Dowec 6 MW pre-design. Aero-elastic modelling of the Dowec 6 MW pre-design in PHATAS.  DOWEC-F1W2-HJK-01-046/9 public version.

[2] Somers, D.M. Design and Experimental Results for the S827 Airfoil. NREL/SR-500-36345 January 2005

---

## Author Comment (AC3)

**Author response to comments by Reviewer 2 of "Local Correlation-based Transition Models for High-Reynolds-Number Wind Turbine Airfoils"**

Yong Su Jung, Ganesh Vijayakumar, James Baeder and Shreyas Ananthan

1 July 2021

We thank Reviewer 2 for the constructive comments and suggestions which have helped to improve the manuscript. We have tried to address most of the concerns as best as possible. We hope that Reviewer 2 would be satisfied by our changes to the manuscript and our responses. Each issue raised by a specific comment in the report is addressed in detail below.

*I appreciate the effort the authors have made to improve the manuscript. It looks (and reads) much better now. However, I still have some doubts about the use of the NREL 5MW data for the comparisons. These data are not only synthesized but also (as the text in the files states) corrected for a blade aspect ratio of 17. Hence the 17 in the file names. The data should be uncorrected for this before they can be properly compared to 2dsimulations. Because of the nature of these data the entry experiment in the legend of the graphs with the comparisons is also not valid. I still wonder why the authors, despite the uncertainties around these data, decide to keep these comparisons in the manuscript.*

We understand the reviewer's concern regarding the reference data for the comparison of DU airfoil series. The main goal of this study is to evaluate the suitability of two exsiting transition models for airfoil design which requires the accurate prediction of the glide ratio near the design operating point region. For DU series airfoils and NACA64-618, the predictions of glide ratio are clearly improved using the two-equation model at the linear portion of the lift curve as shown in Fig. 11 and Fig. A1. The same trend is also confirmed through DU-00-W212 airfoil and FFA series airfoils by comparing with experimental and other predictions using $e^N$ method at various Reynolds numbers.

Although the reference data for DU series airfoils are not only "synthesized" but also corrected for a blade aspect ratio of 17, the current comparisons are still relevant to the main focus of this paper. Also, we believe the difference between the two transition model predictions of the glide ratio is much larger than the effect of aspect ratio because a similar trend is also observed in the other comparison studies for DU-00-W212 and FFA series airfoils. Thus, we would like to keep the section 4.3 for DU series airfoil and NACA64-618 after providing additional information regarding the reference because it can still support the high-level message of this study.

In the revised manuscript, we provided additional informaion regarding the reference data at the section 4.3 as follows:

"Also, the available reference data was corrected for a blade aspect ratio of 17 in the DOWEC 6MW pre-design report [1]. However, we believe the data is still valid as a reference in explaining any differences of model predictions."

**Bibliography**

[1] Kooijman, H. J. T., Lindenburg, C., Winkelaar, D., and van der Hooft, E. L.: Aero-elastic modelling of the DOWEC 6 MW pre-design in PHATAS, Tech. rep., ECN, 2003.

---

## Author Response (AR1)

**Author response to comments by Reviewer 1 of "Local Correlation-based Transition Models for High-Reynolds-Number Wind Turbine Airfoils"**

Yong Su Jung, Ganesh Vijayakumar, James Baeder and Shreyas Ananthan

25 May 2021

We thank Reviewer 1 for the many constructive comments and suggestions which have helped to improve the manuscript. We have tried to address most of the concerns as best as possible. We hope that Reviewer 1 would be satisfied by our changes to the manuscript and our responses.

Each issue raised by a specific comment in the report is addressed in detail below. Modifications of the manuscript can be tracked in the highlighted version of the revised article (red = removed, blue = added or modified).

Line 33-34: I am not sure where in the paper of Sorensen et al. 2016 this is stated. The main goal of that study was to make sure the codes produce consistent solutions in terms of domain size, grid resolution, convergence criteria etc. Moreover, there were no experimental results to compare. Another look at the literature is required to support this statement.

We agree with that the reference is incorrect for the explaining the limitation of 2D RANS-CFD in stall prediction. In the revised manuscript, the reference is corrected to "Ceyhan et al. 2017b".

Line 34-35: accurate prediction of the glide ratio near design points. Line 35: what/where is the design operating point? Normally, the airfoils are designed for a range of angle of attacks not for a single point.

We agree with the suggestions and corrected the revised manuscript as shown below:

"Airfoils are typically designed to operate inside a range of angles of attack for maximum performance away from stall in the linear portion of the lift curve. Hence, the generation of training data for airfoil-design purposes requires the accurate prediction of the glide ratio inside the design range of angles of attack."

Line 35: Please check cross-reference formats throughout the paper. Is it Sorensen (2014) or (Sorensen 2014)? I won't point out all of them but please check and correct the whole paper for this.

We corrected reference format consistently overall the revised manuscript. We follow the rule of in-text citation from the template. Thus, if the reference authors name is part of the sentence structure, only the year is put in parentheses. Otherwise, name and year are put in parentheses. Line 35-36-37: In this paper of Sorensen, there were only three codes with transition model and all of them used eN method. Two of them provided results with different grid resolutions. Although it can be seen there is an effect of the eN transition prediction in the glide ratio and trends with Re numbers, I don't think this is a sufficient evidence for this statement; especially since in that paper there is no experimental results to compare. How about the other transition models used in the literature? For example, as you have also pointed out, Sorensen 2014 showed that the correlation based transition model of Menter was giving wrong trends wrt increasing Re number. There is also another stability theory based transition model used in the paper from Coder. How about this one? It is recommended to check these statements in this paragraph and support with more evidence.

The paper by "Sorensen et al., 2016" only includes the  $e^{N}$  transition results from DTU, NTUA, CENER-WMB codes at 3 and 15 millions and the statement as "Results are not included for the correlation based transition model by Menter and Langtry [15], [16], available in some of the codes, as it fails to correctly predict the natural transition behaviors at high Reynolds numbers."

We agree with the comments from the reviewer, and these are insufficient evidence for the current statement especially in the word selection "transition model". To clarify the paragraph, the sentences in the revised manuscript are changed as shown below:

"Hence, the generation of training data for airfoil-design purposes requires the accurate prediction of the glide ratio inside the design range of angles of attack. The variation of the glide ratio near the design points is highly sensitive to the boundary layer transition onset location."

Also, the stability theory based transition model by Coder (AFT model) is not the subject of this work. We use it as a reference for comparison to state of the art models in Section 3 for validation at low Reynolds numbers where all of the test transition models work well.

**Line 40: "is difficult" - why it is difficult? Do you mean it takes longer to converge? Or something else? Please clarify**

The  $e^{N}$  based approach uses integral boundary layer (IBL) method which solve for quantities not readily available in general CFD approaches. Extracting the associated IBL quantities, such as displacement thickness, momentum thickness, and shape factor, requires nonlocal search and line integration operations for the CFD. To clarify the meaning, we added more explanation in the revised manuscript as shown below:

"However, the application of the  $e^{N}$  method within a conventional RANS framework that runs on massively parallel computers is difficult. This is because it involves non-local search and line integration operations for boundary layer quantities (e.g. displacement/momentum thickness and shape factor)."

**Line 41-42: what is wrong with computing N from semi-empirical models or from a stability solver? Please clarify.**

The semi-empirical model cannot guarantee model accuracy for any arbitrary flow conditions because it is based on a limited experimental dataset. Also, solving a linear stability solver to compute N factor requires additional effort in  $e^{N}$  based model. Additionally, the output of  $e^{N}$  based method should be fed into CFD computation several times until the solutions are fully converged. For better clarification, we changed the sentence in the revised manuscript as shown below:

"Also, additional efforts in communications between  $e^{N}$  and RANS methods are required [7]."

**Line 44: ... in coupling it with ...**

We agree with the suggestions and corrected it in the revised manuscript.

**Line 45-46: How about the suitability of this method? What is right/wrong with it? Was it used for the wind turbine airfoils applications?**

The AFT model is not the subject of this work and it is not used as a reference for comparisons (e.g. LCTM or  $e^{N}$  models) at high Reynolds number flows in this paper. But, in the introduction, we just wanted to cite the model as another model available in the literature that is coupled to the SA turbulence model. As a reference model predicting for S809 airfoil at the low Reynolds number flow, the model is briefly introduced at Section 3 in the revised manuscript as shown below:

"AFT2019 transition model was developed based on linear stability theory, which is also widely used in aerospace problems. It solves two transport equations for amplification factor and intermittency."

Therefore, we chose to delete the current sentence regarding AFT model here.

**Line 74: how about the suitability of the method for the machine learning process as mentioned in the beginning as the goal of the study?**

The machine-learning approach to airfoil design will only use the polar data generated by the CFD solver and hence is independent of and agnostic to the choice of the transition model. This approach will only require accurate prediction of the quantities of interest relevant to design just like any other design approach.

**Line 125 - paragraph. This paragraph seems incomplete. what is ysep and Gonset? Why these are not needed? What is the advantage or disadvantage?**

In the manuscript, the  $\gamma_s$  and  $G_{onset}$  are the variables for explaining main differences between current two-equation implementation and Langtry-Menter (2009) models. However, the details of the differences between the models are already explained in the previous study which is already cited as [5]. We think detailed explanations of the differences are redundant in this manuscript. Thus, we changed the statements in the revised manuscript as shown below:

"Details of the current implementation of the transition model compared to  $\gamma - \overline{Re_{\theta t}}$ model by Langtry and Menter (2009) are shown in the previous study (Medida, 2014; Jung and Baeder, 2019)." General question about the implementation of the two eqn. transition model: was there a need to modify the original correlations or a new calibration to obtain better results for high Re number cases? Should we expect any influence?

The two-equation model retains the primary features of the Lantry-Menter model (2009). However, the following changes were made by Medida et al. (2014):

- 1. New correlation for  $Re_{\theta t}$ ,
- 2. Constant freestream turbulence intensity,
- 3. Modified production and destruction terms in the intermittency equation,
- 4. Omission of the separation-induced transition modification,
- 5. Destruction term in the SA model not scaled by intermittency.

We use the implementation and correlations from Medida et al. (2014). No additional modification or new calibration especially for high Reynolds numbers are made in this paper.

Line 164: Why start with "However"? If the turbulence intensity is not a variable in the SA turbulence model and the current study is also assumes that the measured turbulence intensity is constant everywhere, this should be convenient for the SA turbulence model right?

We agree with the comment. Thus, in the revised manuscript, the sentences are changed to prevent any possible confusion.

Line 170: Validation. This section is hard to follow. A few subsections would help the reader to find its way easier.

Based on this and another similar suggestion from Reviewer 2, we have reorganized sections 3 and 4 into validation of the turbulence model and the transition models. Section 3 now contains only validation results for the SA turbulence model using the fully-turbulent flow assumption, while Section 4 contains the validation results for all simulations using the transition model. This is only reflected in the output of *latexdiff* program for the response to Reviewer 2.

Line 189-190: "... This implementation of the SA model..." this sentence seem redundant. We already know this.

We agree with the suggestions and deleted the sentence in the revised manuscript.

Line 183: What can you say about the comparisons about the grid resolution and other parameters used?

Both simulations used enough fine meshes for the fully turbulent flow simulations, thus the both predictions have minor mesh dependency. From the reference (Bak et al., 2013), the enough fine resolution mesh was used for EllipSys2D to ensure mesh independence; 512 cells around the airfoil and initial wall normal spacing of  $5 \times 10^{-7}$  chord. In the current simulation, 400 points around the airfoil and initial wall normal spacing of  $2 \times 10^{-6}$  chord which correspond to  $y^+ = 1$  were used as already discussed in the Methodology section. Also, the minor mesh dependency in the current study is shown in Appendix B in the revised manuscript.

In terms of the other solver parameter, both simulations neglected any compressibility effects. In the current study, a freestream Mach number is set as 0.1 to represent incompressible flow condition. Otherwise, EllipSys2D is an incompressible solver. In the revised manuscript, the sentences are added as shown below:

"Both predictions used enough fine meshes for the fully turbulent flow simulation, thus there is minor mesh dependency on both predictions. Also, both simulations neglected compressibility because EllipSys2D is a incompressible solver."

**Figure 3: Legend should be HAM2D and not Fully turb**

The legend is fixed as HAM2D in the revised manuscript

Line 191: in this paragraph it is not clear whether you run Overflow yourself or you used the data from the transition modeling workshop.

In this study, we did not run OVERFLOW ourselves. Instead, we used data from the previous studies (Coder, 2019; Hall, 2018). For better clarification, the sentence is corrected in the revised manuscript as shown below:

"We show validation of the aerodynamic performance prediction against experimental data [8] as well as previous simulation results using NASA's OVERFLOW code from Coder [2] using SA-neg turbulence model with AFT2019 transition model."

**Line 194: What kind of transition model is this AFT2019?**

AFT2019 model is based on linear stability theory rather than local correlations is coupled with the SA turbulence model. In general CFD approaches, it solves two transport equations for amplification factor (n) and intermittency ( $\gamma$ ). The amplification factor transport equation was originally derived based on Drela-Giles model [3]. For more details, the sentence is added in the revised manuscript as shown below:

"AFT2019 transition model was developed based on linear stability theory, which is also widely used in aerospace problems. It solves two transport equations for amplification factor and intermittency."

Line 195: "... Test flow condition is at free stream..." I don't understand this sentence. What is two equation Mach number?

The typo is corrected in the revised manuscript.

Line 202: Do you also use the simulation results from Coder 2019 paper? As stated above it would be good to clarify which simulation comes from where and be consistent with how you refer to it in different places in the paper.

We used OVERFLOW simulation results from previous studies (Coder, 2019; Hall, 2018). We clarified the source of the results in the sentences for the revised manuscript.

Line 204: What are the "known limitations of 2D CFD RANS"? How about whether or not these codes were able to resolve the laminar separation bubble at these simulations? Experimental results from TU Delft Wind tunnel as used by Somers https://www.nrel.gov/docs/legosti/old/6918.pd show this bubble clearly and how lift actually continues to rise once post stall after the bubble is gone. I would expect a more elaboration on this physical phenomenon and how these codes were/were not able to capture this.

While there are several "known limitations of 2D CFD-RANS", we are specifically referring to the over-prediction of the stall angle for airfoils in this sentence. The current two-equation model capability for capturing the laminar separation bubble at mid-chord was already validated in a previous study by Jung and Baeder [5].

As shown in Fig. RC1-1, the surface pressure distribution are plotted using different resolution meshes and the predictions are compared to experimental data [8] at 1, 6, and 9° angles of attack. As an indicator of the laminar separation bubble, a small region of flattened surface pressure is well observed, especially on the lower surface in both experiment and the predictions.

Figure RC1-1: Surface pressure distribution for S809 airfoil at  $Re = 2 \times 10^6$  from Jung and Baeder [5].

Line 206: Again in this paragraph, please make sure you refer to the correct references and address where the simulations come from consistently.

We revised the manuscript as shown below:

"Figure 4 (b) shows that the drag predictions from HAM2D using the fully turbulent approximation are in excellent agreement with the OVERFLOW simulation results (Coder, 2019) at the same flow condition over the full range of angle of attack while showing a slight underprediction in the drag bucket compared to the trippped boundary layer experimental data."

Figure 4a: Is the experimental results are for the tripped case? I think both tripped and untripped Cl should be available. Although the difference was small in the case of TU Delft experiments. How about the Cl results for the negative angle of attack values?

---

## Referee Report (RR1)

**Review of WES-2021-23 revised manuscript**

I would like to thank authors for the effort to improve the paper with the suggestions. It looks much better and now it is easier to follow. However, there is one particular issue which I would like to highlight here.

The value of using synthesized RFOIL simulation results of NREL 5MW airfoils as a "validation case" is still not really clear in the manuscript. Even though you mention that it is for the sake of explaining the differences of one and two eqn. transition model predictions, you still treat the reference data as if it is the correct one. Example: line 354-355 "All simulations miss the prediction of stall angle as is typical of the challenges in 2D RANS-CFD modeling of airfoils". The accuracy of the stall angle in the reference data is surely questionable because it is also coming from a prediction method. Furthermore, in Figure 11, this data is still referred as "experimental" and at least the drag data of both airfoils higher than about 10 degrees of angle of attack seem doubtful as if they are wrongly synthesized or the interpolation or correction of the high AR effects are not done correctly perhaps. These kind of issues bring further doubts about the validity of the reference; hence the value and the validity of the comparisons.

I recommend to revise the manuscript further to clarify these issues.

---

## Editor Decision (ED1)

Lines 315-317 state:

"Our understanding is that the reference data for the DU airfoils (Timmer, 2021) were "synthesized" using RFOIL (Van Rooij, 1996) for the Reynolds number of 7 million using correction factors on the basis of a comparison of RFOIL calculations and measurements at 3 million from the Delft wind tunnel in the clean configuration."

I am afraid that this sentence adds more mist to the understanding of the situation.

The point is that the Re= $3 \times 10^6$ data for the DU airfoils (figures 10 top three) are the result of experiments in the LTT wind tunnel of TU Delft. The post stall behaviour is indeed the behaviour "seen" in the wind tunnel. The "jumpy" behaviour is the result of a "non 2 dimensional flow" that typically occurs in the post stall area. So the measured values are correct for the measured span wise location, but will differ when the measurements would have been done at a different span wise location.

The bottom three figures of figure 10 are the airfoil data for Re=$12 \times 10^6$. Such Re numbers cannot be realised in the LTT wind tunnel of TU Delft and thus are not the direct result of experiments. They are synthetized from the Re= 3x106 data using the airfoil design code RFOIL.

So I would suggest to replace the above sentence into:

"The reference data for the DU airfoils at Re=$3 \times 10^6$ are taken from experiments in the LTT wind tunnel of TU Delft. The results for the higher Re numbers (typically Re=$6 \times 10^6$, Re=$7 \times 10^6$ and Re=$12 \times 10^6$) are the result of a synthesis process, in which measured data for at Re=$3 \times 10^6$ are translated to higher Re numbers using the airfoil design code RFOIL (Van Rooij 1996)."

And consequently you should change the legend in the lower three graphs of figure 10 from "experiment" to "reference".

---

## Author Response (AR2)

**Author response to comments by Referee 1 of "Local Correlation-based Transition Models for High-Reynolds-Number Wind Turbine Airfoils"**

Yong Su Jung, Ganesh Vijayakumar, James Baeder and Shreyas Ananthan

23 November 2021

We thank Reviewer 1 for the comments. Again, we have tried to address the reviewer's concern and revised the manuscript based on the comments. We hope that Reviewer 1 would be satisfied by our responses and following changes to the manuscript.

*I would like to thank authors for the effort to improve the paper with the suggestions. It looks much better and now it is easier to follow. However, there is one particular issue which I would like to highlight here. The value of using synthesized RFOIL simulation results of NREL 5MW airfoils as a "validation case" is still not really clear in the manuscript. Even though you mention that it is for the sake of explaining the differences of one and two eqn. transition model predictions, you still treat the reference data as if it is the correct one. Example: line 354-355 "All simulations miss the prediction of stall angle as is typical of the challenges in 2D RANS-CFD modeling of airfoils". The accuracy of the stall angle in the reference data is surely questionable because it is also coming from a prediction method. Furthermore, in Figure 11, this data is still referred as "experimental" and at least the drag data of both airfoils higher than about 10 degrees of angle of attack seem doubtful as if they are wrongly synthesized or the interpolation or correction of the high AR effects are not done correctly perhaps. These kind of issues bring further doubts about the validity of the reference; hence the value and the validity of the comparisons. I recommend to revise the manuscript further to clarify these issues.*

- We agrees with the reviewer's comment on "experiment" as a legend of the reference data in Fig. 11. Therefore, we have changed the legend of the reference data from "Experiment" to "Reference" in Fig. 11. and Fig. A1. In the manuscript, the information on how the reference data were generated is already provided in detail from the previous revision in line 315-320.
- Reviewer doubts about the validity of the reference data by pointing the drag data at higher than 10 degree angles of attack, where the drag is sharply increased linearly. However, a similar behavior can be seen in the experimental data for DU00-W-212 airfoil in Fig. 10. The experimental data shows earlier stall angle than the corresponding predictions at both Reynolds numbers. More like linear shape of drag after stall in the reference data in Fig. 11 might be due to the synthesized or interpolation process as the reviewer already commented. However, the data after stall is not the main subject of the current study and cannot be an issue.

- Reviewer concerns about the accuracy of the stall angle of the reference data in Fig. 11 because it is also coming from a prediction method (RFOIL) partially. However, when it is compared with the current predictions, the trends of differences are close to the comparisons with experiments as shown in Fig. 10. On the other hand, as another reference in the manuscript, the pure predictions using EllipSys2D shows the same stall behavior with one of current predictions as shown in Fig. 12. This might be due to either/both effects of the synthesizing experiment or/and better stall prediction using RFOIL. Therefore, we think the synthesized data is still valid as a "Reference" not an "Experiment".
- In the revised manuscript, therefore, we have updated the corresponding discussion for Fig. 11 as follows:
  "Overall, the current simulations predicted delayed stall angles compared to the reference data. It should be noted that the same trend is also observed in the previous comparison with pure experimental data for the DU00-W-212 airfoil in Fig. 10, which is a typical challenge in 2D RANS-CFD modeling of airfoils. The unphysical linear increment in the drag coefficient is observed after the stall angle only in the reference data. This might be due to the synthesized process between RFOIL calculations and experimental data."

---

## Author Response (AR3)

**Author response to comments by Associate Editor of "Local Correlation-based Transition Models for High-Reynolds-Number Wind Turbine Airfoils"**

Yong Su Jung, Ganesh Vijayakumar, James Baeder and Shreyas Ananthan

24 January 2022

We thank Associate Editor for the comments. We hope that Associate Editor would be satisfied by our changes to the manuscript our responses.

First, We would likt to point that the source of reference data between Fig. 10 and 11 are different. The current manuscript line 235-245 explains the source of the refrence data for the Fig. 10, which is from experiment without the synthesis process. In the Fig. 10, the reference data for the DU00-W-212 airfoil were taken from DNW-HDG wind tunnel in Gottingen within AVATAR project, not from LTT wind tunnel in TU Delft as mentioned in the references as below [1, 2].

"The DU00-W-210 airoil was tested in the DNW-HDG pressurized wind tunnel in order to investigate the flow at high Reynolds number range from 3 to 15 million which is the operating condition of the future large 10MW+ offshore wind turbine rotors."

The lines 315-317 which are mentioned by Associated Editor is only about Fig. 11 were replaced as suggested in the revised manuscript as below.

"The reference data for the DU airfoils at $Re = 3 \times 10^6$ are taken from experiments in the LTT wind tunnel of TU Delft. The results for the $Re = 7 \times 10^6$ are the result of a synthesis process, in which measured data for at $Re = 3 \times 10^6$ are translated to the higher Reynolds number using the airfoil design code RFOIL [3]."

**Bibliography**

[1] Ceyhan, O., Pires, O., and Munduate, X.: AVATAR HIGH REYNOLDS NUMBER TESTS ON AIRFOIL DU00-W-212, Tech. rep., https://doi.org/10.5281/zenodo.439827, URL https://doi.org/10.5281/zenodo.439827, 2017.

[2] Ceyhan, O., Pires, O., Munduate, X., Sorensen, N., Reichstein, T., Schaffarczyk, A., Diakakis, K., G, P., Daniele, E., M, S., Lutz, T., and Prieto, R.: Summary of the Blind Test Compaign to predict the High Reynolds number performance of DU00-W-210 airfoil, in: AIAA Scitech, 2017.

[3] Van Rooij, R.: Modification of the boundary layer calculation in RFOIL for improved airfoil stall prediction, 1996.